# Mapping variants in thyroid hormone transporter MCT8 to disease severity by genomic, phenotypic, functional, structural and deep learning integration

Predicting and quantifying phenotypic consequences of genetic variants in rare disorders is a major challenge, particularly pertinent for 'actionable' genes such as thyroid hormone transporter MCT8 (encoded by the X-linked *SLC16A2* gene), where loss-of-function (LoF) variants cause a rare neurodevelopmental and (treatable) metabolic disorder in males. The combination of deep phenotyping data with functional and computational tests and with outcomes in population cohorts, enabled us to: (i) identify the genetic aetiology of divergent clinical phenotypes of MCT8 deficiency with genotype-phenotype relationships present across survival and 24 out of 32 disease features; (ii) demonstrate a mild phenocopy in ~400,000 individuals with common genetic variants in MCT8; (iii) assess therapeutic effectiveness, which did not differ among LoF-categories; (iv) advance structural insights in normal and mutated MCT8 by delineating seven critical functional domains; (v) create a pathogenicity-severity MCT8 variant classifier that accurately predicted pathogenicity (AUC:0.91) and severity (AUC:0.86) for 8151 variants. Our information-dense mapping provides a generalizable approach to advance multiple dimensions of rare genetic disorders.

A rare disorder is typically defined as occurring in less than 1 in 2000 individuals; with over 5000 known rare genetic disorders, approximately 1 in 20 people are affected globally. Our knowledge of the effects of genetic variants in such disorders is still very limited, signifying an unmet need. Predicting and quantifying the effects of variants on phenotypic manifestation and disease severity in rare disorders is a major undertaking given the limited availability of systematic collection of clinical and functional data. This is particularly important for 'actionable' genes, i.e. genes in which the definitive ascertainment of a pathogenic variant and/or the severity of its impact may optimize individualized clinical management[1]. Moreover, while transmembrane proteins represent up to 30% of the genes in the human genome, less than 0.5% of experimentally determined structures are of integral membrane proteins, limiting the use of such structures as input for the prediction of functional consequences of variants[2-5].

The abovementioned challenges are epitomized by monocarboxylate transporter (MCT)8 deficiency. MCT8 is a transmembrane transporter that is crucial for transport of the thyroid hormones triiodothyronine (T3) and thyroxine (T4) into several tissues, including the brain[6-10]. Pathogenic variants in *SLC16A2*, the gene encoding MCT8 located at the X-chromosome, occur in approximately 1:70,000 male individuals[11-13]. MCT8 deficiency is characterized by a varying neurodevelopmental delay due to cerebral hypothyroidism, and a wide range of clinical sequelae secondary to chronic peripheral thyrotoxicosis caused by elevated serum T3 concentrations[14]. In an international clinical trial we showed that treatment with the T3 analogue tri-iodothyroacetic acid (Triac) ameliorates key (thyrotoxic) features of this disease[15].

✉e-mail: w.e.visser@erasmusmc.nl

In this study, we systematically collected genetic, clinical and biochemical data from individuals with MCT8 deficiency accrued through a world-wide collaboration across 53 sites in 23 countries[14] supplemented with information from all described cases in literature and data from up to 406,975 non-affected individuals. These data were integrated with (i) molecular studies, utilizing different optimized functional assays in both transfected and patient-derived cells[16], (ii) extensive alanine-scanning of the protein, and (iii) in silico approaches including a newly constructed MCT8 homology model, ultimately constructing a machine-learning based dual pathogenicity-severity classifier (Fig. 1A). Here, beyond advancing the understanding of fundamental molecular characteristics of this transporter and providing

**Fig. 1 | Overview of study and characteristics of the study cohort. A** Overview of workflow. (1) Meta-analysis of disease features in patients with MCT8 deficiency; (2) functional analysis of benign and disease-causing variants and mapping functional outcomes to disease manifestations; (3) genotype-phenotype analysis in non-affected populations; (4) evaluation of therapy effectiveness in different disease-severity classes; (5) alanine-scanning and in-depth functional characterization to map critical residues onto the protein structure; (6) disease variant and severity classifier. Dashed blue arrows between boxes indicate input for other packages. Created in BioRender. Visser, E. (2023) BioRender.com/c51c336. **B** Developmental, clinical, imaging and biochemical disease features in a meta-analysis of patients ($n = 371$) with MCT8 deficiency. Abbreviations: MRS, magnetic resonance spectroscopy; WMV, white matter volume; WML, white matter lesions; ALT, alanine aminotransferase; EEG, electroencephalogram; SHBG, sex-hormone binding globulin. **C** Overview of unique genetic mutations identified in the *SLC16A2* gene, encoding MCT8, and investigated in this study. (left panel) deletions (lines) and splice site mutations (arrow heads) and (right panel) missense (circles), nonsense (triangles), indel (squares) and frameshift (diamonds) mutations. Mutations that occurred >1 in independent families are indicated with their frequency inside the symbol. Details of mutations are presented in Supplementary Fig. 3. **D** Schematic overview of the nature and functional impact of all tested missense, nonsense, frameshift and indel MCT8 variants. The inner ring indicates the proportion of the nature of variants (strongly transparent: missense variants; mildly transparent: nonsense variants; hardly transparent: frameshifts; not transparent: indels). The outer ring represents residual T3 transport capacity of all tested MCT8 variants shown as functional impact (0% = wild-type MCT8; 100% = no residual function; mean ± s.e.m.) in COS-1 cells. Data were derived from ≥3 experiments with technical duplicates. Red indicates exon 1; orange indicates exon 2; yellow indicates exon 3; green indicates exon 4; blue indicates exon 5; purple indicates exon 6; * common or rare SNPs in non-patients; # indicates a missense variant that also likely affects splicing.

an integrative atlas informing clinical management and future trials in MCT8 deficiency, our study provides a framework for robust comprehensive mapping of the phenotypic, functional and structural consequences of disease-causing variants in other rare genetic disorders that also face manifold challenges.

## Results

### Meta-analysis of genetic and disease characteristics

We first assembled a comprehensive overview of genetic, developmental, clinical and biochemical disease characteristics through a meta-analysis of 371 patients with MCT8 deficiency (Supplementary Fig. 1). Affected individuals were identified through our international consortium on MCT8 deficiency[14], consisting of our reference cohort ($n = 151$) and newly identified patients ($n = 14$) from 53 sites in 23 countries (Supplementary Fig. 2), and through a systematic literature search, identifying another 206 unique and independent individuals with MCT8 deficiency (see Supplementary Information). Key demographic, developmental, clinical, imaging and biochemical features are summarized in Fig. 1B and Supplementary Tables 1–3, extending our previous data and substantiating the prodigiousness of the dataset[14]. In total 155 different variants in *SLC16A2* were identified in patients with MCT8 deficiency, representing 63 missense variants, 22 nonsense, 37 frameshift variants, 9 deletions or insertions of 1–3 amino acids, 14 large deletions and 10 splicing variants (Fig. 1C and Supplementary Fig. 3). Most hotspots (15 out of 155 unique variants present in ≥3 independent families) occurred in exons 2, 3 and 4, corresponding to transmembrane domains (TMDs) 2, 4 and 8. To identify non-synonymous variants that may cause less extreme or no distinctive phenotypes, we investigated whole-exome sequencing data from 34,057 non-affected individuals and identified 14 non-synonymous variants (1 common, 13 rare) in *SLC16A2* present in at least 2 males (Supplementary Table 4).

### Mapping MCT8 variants to disease manifestations

To enable functional interpretation, we constructed a cDNA library of disease-causing ($n = 112$) and benign ($n = 18$) MCT8 variants (Supplementary Methods). Functional evaluation of these variants in two different cell lines showed that the majority of disease-causing variants completely abolished MCT8-mediated T3 transport (COS-1 cells: 70 out of 112 variants, 62.5%; JEG-3 cells: 76 out of 112 variants, 67.9%) and T4 transport (COS-1 cells: 75 out of 112 variants, 67.9%; JEG-3 cells: 77 out of 112 variants, 68.8%), whereas the remainder retained a variable degree of residual transport capacity (Fig. 1D, Supplementary Fig. 4). About 78% of variants had <70% membrane expression compared to wild-type MCT8 protein (Supplementary Fig. 5). By contrast, all 14 non-synonymous variants identified in population-based studies had normal MCT8 function (Fig. 1D, Supplementary Fig. 4a, Supplementary Table 4).

We next determined whether there was a quantitative relationship between the degree of functional impact (mild, moderate and severe loss of function (LoF)) of the variants and survival, as well as 32 phenotypic outcomes in 358 patients (Supplementary Fig. 1). LoF was similar across the different cell models; as variation in T3 uptake across variants was largest in COS-1 cells, we elected those cells as the primary model to perform genotype-phenotype analyses (Supplementary Fig. 4b). Residual transport capacity predicted survival when patients were stratified across LoF classes (Fig. 2A; Table 1). To ascertain clinical phenotypes associated with the functional impact of variants, we capitalized on key neurological and metabolic outcomes present in our meta-analyzed cohort. Key developmental milestones, seizures (Fig. 2B–E) and other neuropsychological tests and clinical examinations (Table 1; Supplementary Fig. 6) were directly linked with the different LoF classes. Key metabolic and thyrotoxic features, including underweight and premature atrial contractions, were more frequently observed among individuals harboring severe LoF variants (Fig. 2F, G; Supplementary Figs. 7, 8, Table 2). Cardiac arrhythmia and conduction abnormalities were only present in patients with severe or moderate LoF-variants, and absent in those with a mild LoF variant (Table 2). Likewise, serum free T4 and T3 concentrations as well as biochemical markers reflecting thyroid hormone action were increasingly abnormal depending on the degree of functional impairment (Fig. 2H–J and Supplementary Fig. 7, Table 2). We replicated those findings using both T3 and T4 as substrates in the functional assays in JEG-3 cells (Supplementary Fig. 9). Moreover, we substantiated these findings with data obtained in fibroblasts from 34 patients. We noted a pronounced reduction in cellular T3 transport compared to control fibroblasts, which correlated with the overexpression studies, highlighting the relevance of both models (Supplementary Fig. 10). Complete versus partial LoF in patient-derived fibroblasts forebode survival and severity among key disease characteristics (Fig. 2K–M).

### Gene-based common-variation assessment in the general population

With features arising from disease-causing variants representing the extreme end of a spectrum, we next investigated whether common genetic variation in MCT8, as present in the general population, was associated with some phenotypic resemblances of MCT8 deficiency. Therefore, we performed a genetic look-up with two complementary analyses for the *SLC16A2* gene in up to 406,975 individuals for thyroid function, metabolic and brain related traits. Gene-based analysis sets and two single non-exonic variants (rs4892386 and rs67736575) were significantly associated with serum free T4 but not TSH concentrations, with trends for BMI, heart rate and different brain functional and MRI-based morphology outcomes (Fig. 3; Supplementary Fig. 11; Supplementary Table 5).

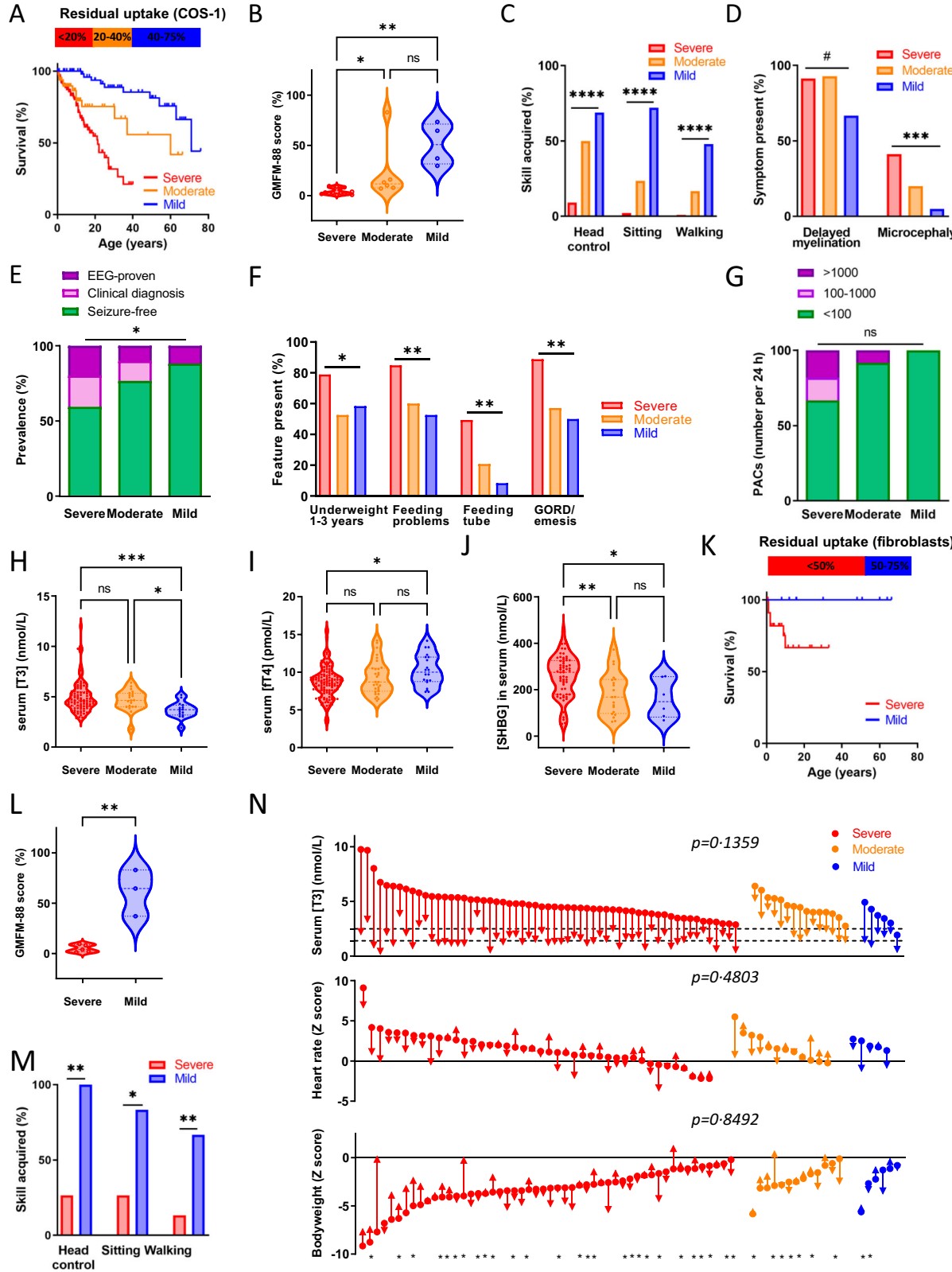

## Therapy effectiveness

We next explored whether therapy effectiveness of Triac[15,17] was similar across 80 patients harboring mutations with different LoF impact. In these patients, Triac treatment according a predefined protocol with a median duration of 22.4 (IQR:12.9–42.4) months and a median dose of 40.5 (IQR:28.9–58.0) μg/kg was equally effective in reducing serum T3 concentrations, as well as improving clinical and biochemical outcomes irrespective of the functional impact of the underlying variant, leveraging the potential impact of treatment on thyrotoxic features to all patients (Table 3, Fig. 2N; Supplementary Fig. 12).

## Molecular characterization

To identify critical residues within MCT8, we performed a systematic Ala-scanning (a technique aiming to determine the contribution of

**Fig. 2 | Phenotypic and treatment outcomes in MCT8 deficiency linked to transport capacity of pathogenic MCT8 variants.** Phenotypic and treatment outcomes in MCT8 deficiency linked to transport capacity of pathogenic variants in transfected COS-1 cells (**A**–**J** and **O**), or patient-derived fibroblasts (**K**–**M**) Patients with MCT8 deficiency are stratified across different LoF classes of functional impact: residual T3 transport capacity in COS-1 cells: <20%, severe LoF (red); 20–40%, moderate LoF (orange), 40–75%, mild LoF (blue), and residual T3 transport capacity in patient-derived fibroblasts: <50%, severe LoF (red); 50–75%, mild LoF (blue). **A, K** Overall survival based on age at last follow-up (Kaplan-Meier estimates). **B, L** Gross Motor Function Measure (GMFM)-88, where 100% corresponds to motor development of a healthy 4-years-old child. **C, M** Ascertainment of developmental milestones. **D** Brain outcomes. **E** EEG-proven or clinical suspected seizures. **F** Clinical features of feeding status. **G** Premature atrial contractions (PACs) during 24 h cardiac monitoring. **H**–**J** Biochemical measurements. Serum concentrations of (**H**) T3 and (**I**) FT4 and (**J**) sex hormone binding globulin (SHBG). **N** Disease outcomes in patients treated with Triac. Five of the 85 patients with available treatment data (Supplementary Fig. 1) were excluded from the analyses as they harbored a splice-site variant. Changes from baseline to last available follow-up visit in serum concentrations of T3 (upper), heart rate-for-age (middle) and bodyweight-for-age (bottom). Median treatment duration was 22.4 months (range 2.0 – 74.5 months). * denotes patients with less than 2 years of treatment. Dashed lines represent reference intervals. For **B**–**M** statistically significant differences between groups are denoted by * $P < 0.05$, ** $P < 0.01$, *** $P < 0.001$, **** $P < 0.0001$, # $P < 0.1$ with a Chi-squared test (**C**–**G**, **M**), Kruskal-Wallis test followed by Dunn's multiple comparisons test (**B**, **H**), unpaired t-test (**L**), one-way ANOVA followed by Tukey's multiple comparisons test (**I**, **J**) and one-way ANOVA (**N**). Exact P values are provided in Supplementary Table 9. Source data are provided as a Source Data file.

specific residues to a protein's function) covering 375 residues from the start of TMD1 until the intracellular C-terminus (Fig. 4A). Up to 23% of Ala variants had residual T3 and T4 uptake capacity of <70% compared to wild-type (Fig. 4A; Supplementary Fig. 13a). Such variants affected residues that are highly conserved across species and within the MCT-protein family (Supplementary Fig. 14), and align with those residues that have previously been identified as critical residues in other MCTs (Supplementary Fig. 15).

Mapping the Ala variants onto revised MCT8 homology models in inward-open and outward-open configurations (Supplementary Figs. 13b, 16–20; Supplementary Information) allowed for the identification of seven critical regions in the protein (Fig. 4B; Supplementary Figs. 21–26; Supplementary Information). Apart from residues making up, supporting or flanking the substrate binding center (TMD1, 4, 7, 10), we identified two critical clusters (TMD5/8 and TMD2/11, representing the rocker-helices[18]) that determine the relative position of the N- and C-terminal halves of the protein, connected by a linker (second half of TMD4/ICL2) (Fig. 4C, Supplementary Figs. 21–26 and Supplementary Information). We then interrogated the Ala-scanning dataset to identify residue(s) that determine(s) substrate specificity. We found N193 as having a pronounced differential impact on T3 versus T4 transport (Fig. 4D). Homology modeling indicated that N193, which aligns to the substrate-interacting K38 in hMCT1, may form a transient halogen bond with the 5'-iodine at the phenolic-hydroxyl group of T4, which is absent in T3 (Fig. 4E). Substantiating this explanation, N193A showed a strong reduction in 3,3',5'-triiodothyronine (containing the 5'-iodine) transport, but a relatively preserved transport of 3,3'-diiodothyronine (lacking the 5'-iodine) (Fig. 4F, Supplementary Fig. 21). In 33 out of 61 missense variants identified in patients, the Ala variant at the affected position was also damaging, implying a critical role for the original residue (e.g. F298, L304, Y354, R445, D498) (Fig. 5B–D, Supplementary Figs. 5, 13c, Supplementary Table 6). Indeed, substitution of these residues by a residue with similar structural and/or chemical properties (partly) restored transport function and/or protein stability (Fig. 5B–D, Supplementary Figs. 5, 13d, e), which allowed to deduce critical local side-chain properties (Supplementary Table 6). As a corollary, other missense mutations occurred at residues of which the Ala-substituent was well-tolerated, pointing to a damaging effect of the substituent rather than the loss of the native residue. Indeed, 26 out of 28 of these variants (e.g. C283Y, S290F, G401R, G564R) reduce protein expression levels and/or interfere with cell membrane translocation (Fig. 5A, D, Supplementary Fig. 5). Only 11 out of the 67 different missense variants, identified in patients with MCT8 deficiency, affected residues in TMD3,6,9,12, which are predicted to face the lipid bilayer and contribute little to the seven identified critical regions (Fig. 5C). Notably, such variants had a marginal impact on MCT8 function.

### Deep learning-based classifier of variant severity

Finally, taking a semi-supervised approach towards building a pathogenicity and severity classifier, we first modelled sequence variation across the tree of life to obtain a continuous pathogenicity score (EVE-score)[19], which was tested against the functional severity data (Fig. 6A–D, Supplementary Fig. 27, Supplementary Table 7). This unsupervised approach was further refined by integrating additional functional data, including the Ala-scan, and structural information via supervised learning aiming to discriminate benign from disease-causing variants (pathogenicity classifier), and within the latter to discriminate severity (severity classifier) (Fig. 6A; Supplementary Methods). The resulting MCT8-specific pathogenicity classifier was found to discriminate benign from disease-causing variants with an accuracy of $0.89 \pm 0.04$ and an area under the receiving operator curve (AUC) of $0.91 \pm 0.08$ estimated by 10-fold cross-validation, outperforming our unsupervised approach (EVE-score) alone (AUC 0.87; $p = 0.001197$), as well as established predictors, such as PolyPhen2 (AUC 0.85; $p = 0.0001515$) and SIFT (AUC 0.71; $p < 0.0001$) and recent state-of-the-art predictor ESM-1v (0.87; $p = 0.006161$) (Fig. 6C). Although our model did not significantly outperform AlphaMissense as measured by AUC (AUC 0.9; $p = 0.27$), we noticed a substantial difference in terms of precision-recall, as indicated by the AUPRC (0.82 vs 0.74, Supplementary Fig. 28a). Next, an MCT8-specific severity classifier, which segregates pathogenic variants into mild or combined moderate and severe LoF, was developed with an accuracy of $0.78 \pm 0.14$ (f1-score $0.80 \pm 0.14$) and an AUC of $0.86 \pm 0.12$ estimated by 10-fold cross-validation, outperforming predictions based on the continuous EVE-score alone (AUC 0.70), and other state-of-the-art predictor including AlphaMissense and ESM-1v (Fig. 6D, Supplementary Fig. 28b). Integrating both classifiers into a dual pathogenicity-severity MCT8 classifier, we predicted the impact of 8151 missense variants into 5183 benign variants and 2987 pathogenic variants (1071 mild LoF and 1916 moderate/severe LoF). Validation experiments with 11 novel patient variants and 6 random artificial variants confirmed the high accuracy of dual pathogenicity-severity MCT8 classifier (Fig. 6E), again outperforming common prediction tools (Supplementary Fig. 29), and satisfactory accuracy in discriminating different LoF categories (Fig. 6F). Finally, we generated a mutational landscape heat map, predicting pathogenicity and severity of all potential missense variants in MCT8 (Fig. 6G and Supplementary Figs. 30, 31).

## Discussion

Here we combined deep phenotyping data from individuals with a rare, multisystem disorder due to MCT8 deficiency with a battery of in vitro functional and computational tests and with outcomes in population cohorts to understand the divergent clinical phenotypes of MCT8 deficiency, assess therapy effectiveness and advance structural and functional insights of this transporter protein. This approach allowed us to construct a high-quality dual pathogenicity-severity variant classifier and also buttressed a role of MCT8 in non-affected individuals in the population.

New genomic technologies have revolutionized the study and care of genetic disorders, impacting the daily lives of individual

**Table 1 | Phenotype (development and neurology) – Loss-of-function correlations**

| | Severe loss-of-function N = 216 | | Moderate loss-of-function N = 50 | | | Mild loss-of-function N = 63 | | |
|---|---|---|---|---|---|---|---|---|
| | | *N* | | *N* | *p* value | | *N* | *p* value |
| Age at assessment (years) | 3.7 (0.3–40.0) | 172 | 6.7 (0.6–64.0) | 36 | | 26.0 (0.9–76.0) | 50 | |
| Survival (years) | 21.4 (17.0–25.8) | 216 | 60 (9.3–110.7) | 50 | 0.024 | 71 (57.1–84.9) | 63 | <0.0001 |
| **Development** | | | | | | | | |
| Head control[a] | 9.1% | 88 | 50.0% | 22 | | 68.8% | 32 | <0.0001 |
| Speech (at least 1 word)[a] | 4.1% | 98 | 29.2% | 24 | | 61.2% | 49 | <0.0001 |
| Independent sitting[a] | 2.1% | 94 | 23.5% | 17 | | 72.0% | 25 | <0.0001 |
| Independent walking[a] | 1.0% | 100 | 16.7% | 24 | | 47.9% | 48 | <0.0001 |
| Total GMFM-88 score (%)[b] | 2.8 (1.6–8.2) | 15 | 11.6 (7.6–32.8) | 6 | 0.029 | 50.8 (31.5–71.2) | 4 | 0.0023 |
| BSID III[a] | | | | | | | | |
| *Cognition* | 3.3 (2.3–4.0) | 15 | 3.6 (0.5–4.8) | 5 | 1.00 | 33.0 (33.0–36.0) | 3 | 0.017 |
| *Receptive language* | 7.0 (3.3–11.0) | 15 | 3.3 (1.0–9.5) | 5 | 0.60 | 31.0 (29.0–32.0) | 3 | 0.026 |
| *Expressive language* | 4.6 (4.6–6.0) | 15 | 4.6 (0.5–5.8) | 5 | 0.78 | 24.0 (21.0–30.0) | 3 | 0.019 |
| *Fine motor skills* | 3.0 (1.0–3.5) | 15 | 3.0 (1.0–4.5) | 5 | 1.00 | 38.0 (25.0–42.0) | 3 | 0.017 |
| *Gross motor skills* | 1.0 (0.5–2.3) | 15 | 1.0 (0.25–1.15) | 5 | 0.83 | 13.0 (9.0–36.0) | 3 | 0.021 |
| VABs II | | | | | | | | |
| *Receptive language* | 12 (8.0–12.0) | 19 | 9.5 (6.5–13.0) | 12 | 1.00 | 28.5 (13.0–38.0) | 4 | 0.095 |
| *Expressive language* | 12.0 (9.0–13.0) | 19 | 11.5 (8.0–12.8) | 12 | 1.00 | 33.0 (17.3–39.0) | 4 | 0.022 |
| *Written communication[b]* | 0.0 (0.0–0.0) | 15 | 0.0 (0.0–0.0) | 8 | 1.00 | 4.5 (0.0–15.0) | 4 | 0.0018 |
| *Personal care* | 6.0 (4.0–6.0) | 19 | 6.5 (2.0–10.8) | 12 | 0.84 | 31.5 (12.5–49.0) | 4 | 0.0071 |
| *Household[b]* | 0.0 (0.0–0.0) | 15 | 0.0 (0.0–0.0) | 9 | 0.75 | 1.0 (0.0–18.5) | 4 | 0.011 |
| *Community[c]* | 0.0 (0.0–0.0) | 19 | 0.0 (0.0–2.0) | 11 | 0.13 | 3.0 (0.5–18.3) | 4 | 0.0009 |
| *Relationships* | 18.0 (14.0–20.0) | 19 | 17.0 (13.0–22.8) | 12 | 1.00 | 31.5 (19.8–37.3) | 4 | 0.12 |
| *Play* | 4.0 (2.0–8.0) | 19 | 5.0 (3.3–8.8) | 12 | 1.00 | 21.0 (8.3–36.8) | 4 | 0.039 |
| *Coping* | 0.0 (0.0–2.0) | 19 | 0.0 (0.0–2.8) | 12 | 1.00 | 10.0 (7.8–11.5) | 4 | 0.0009 |
| *Gross motor skills* | 2.0 (0.0–4.0) | 19 | 3.5 (1.3–4.8) | 12 | 0.45 | 24.5 (6.0–49.8) | 4 | 0.016 |
| *Fine motor skills* | 1.0 (0.0–1.0) | 19 | 3.5 (1.0–6.0) | 12 | 0.034 | 40.0 (12.3–52.0) | 4 | 0.0005 |
| **Neurological signs** | | | | | | | | |
| Hypotonia | 99.3% | 142 | 100.0% | 32 | | 89.5% | 38 | 0.0012 |
| Primitive reflexes | 93.5% | 46 | 69.2% | 13 | | 100.0% | 6 | 1.00 |
| Muscle hypoplasia | 94.0% | 84 | 87.0% | 23 | | 70.7% | 41 | 0.0016 |
| Dystonia | 93.4% | 106 | 83.3% | 30 | | 62.5% | 40 | <0.0001 |
| Spasticity | 89.8% | 127 | 85.7% | 35 | | 90.2% | 51 | 0.76 |
| Plantar extension response (Babinski sign) | 74.1% | 58 | 68.8% | 16 | | 75.0% | 32 | 0.89 |
| Seizures | | | | | | | | |
| *Clinical diagnosis* | 20.7% | 116 | 13.3% | 30 | | 0.0% | 34 | <0.0001 |
| *EEG proven* | 19.8% | 116 | 10.0% | 30 | | 11.8% | 34 | <0.0001 |
| Delayed myelination (MRI) | 91.3% | 69 | 92.9% | 14 | | 66.7% | 9 | 0.073 |
| Microcephaly | 41.2% | 85 | 20.0% | 25 | | 4.9% | 41 | 0.0001 |
| Head circumference (Z-score) | –1.47 | 69 | –0.94 | 20 | 0.1117 | –0.13 | 36 | <0.0001 |

Data are median (interquartile range[IQR]), n (%), or mean (SD). For age, median (range) is provided. For survival median (95% CI) is provided. Loss-of-function classification is based on residual T3 uptake capacity in COS-1 cells. Please note that most parameters have not been captured in all patients.
[a]Patients younger than 2 years were excluded from analyses.
[b]Patients younger than 4 years were excluded from analyses.
[c]Patients younger than 1 year were excluded from analyses.
For proportional data, Chi-square tests were used and p-values in the last column indicate the presence of statistically significant differences between the LoF categories. In all other cases, One-way ANOVA (or its non-parametric alternative) were used with Tukey's post-tests; indicated are the p-values for comparisons of severe *versus* moderate and severe *versus* mild LoF categories. Source data are provided as a Source Data file.
*BSID* Bayley's Score of Infant Development, *GERD* gastro-oesophageal reflux disease, *GMFM* Gross Motor Function Measure, *LLN* lower limit of normal, *PACs* premature atrial complexes, *VABs* Vineland Adaptive Behavior Scales.

patients, changing paradigms of pathophysiology and paving the way for new avenues in therapy development. However, with phenotypes varying widely among individuals with a specific genetic disorder, rare diseases have daunting hurdles such as disease unawareness, lack of global expertise along with a wide geographical spread which hinders rigorous collection of all-inclusive data. Furthermore, a lack of adequate mutation-based phenotypic prediction and limited structural information, particularly for membrane proteins, contribute to such challenges. These difficulties are epitomized by MCT8 deficiency. Hence, we believe that our global,

**Table 2 | Phenotype (Thyrotoxicosis and biochemical measurements) – Loss-of-function correlations**

| | Severe loss-of-function N = 139 | | Moderate loss-of-function N = 32 | | | Mild loss-of-function N = 34 | | |
|---|---|---|---|---|---|---|---|---|
| | | N | | N | p value | | N | p value |
| **Thyrotoxic signs** | | | | | | | | |
| Underweight | 72.8% | 139 | 50.0% | 32 | | 52.2% | 23 | 0.0055 |
| Weight-for-age (Z-score) | −3.05 | 118 | −2.00 | 25 | 0.0124 | −2.25 | 18 | 0.1134 |
| Short stature | 38.8% | 112 | 27.6% | 29 | | 11.8% | 34 | 0.0036 |
| Height-for-age (Z-score) | −1.86 | 101 | −1.13 | 23 | 0.0901 | −0.87 | 29 | 0.0106 |
| Feeding status | | | | | | | | |
| *Feeding problems* | 84.8% | 99 | 60.0% | 25 | | 52.6% | 19 | 0.0012 |
| *Tube feeding* | 49.4% | 79 | 20.8% | 24 | | 8.3% | 12 | 0.0030 |
| *GERD* | 88.9% | 45 | 57.1% | 14 | | 50.0% | 8 | 0.0062 |
| *Underweight (age 1–3 yrs)* | 78.9% | 90 | 52.6% | 19 | | 58.3% | 12 | 0.033 |
| Heart rate | | | | | | | | |
| *Heart rate (Z-score)* | 1.16 (2.01) | 68 | 1.48 (1.57) | 16 | 0.47 | 2.11 (0.56) | 6 | 0.77 |
| *Tachycardia in rest* | 30.3% | 66 | 26.3% | 19 | | 42.9% | 7 | 0.48 |
| >100 PACs (per 24 h) | 33.3% | 33 | 8.3% | 12 | | 0.0% | 5 | 0.092 |
| Other cardiac signs[a] | | | | | | | | |
| *Arrhythmia* | 18.5% | 27 | 7.7% | 13 | | 0.0% | 5 | 0.42 |
| *Abnormal conduction* | 20.7% | 29 | 23.1% | 13 | | 0.0% | 5 | 0.51 |
| *Prolonged QTc* | 4.2% | 24 | 11.1% | 9 | | 20% | 5 | 0.45 |
| QTc (s) | 0.417 | 25 | 0.424 | 8 | 0.5177 | 0.432 | 4 | 0.2756 |
| Systolic blood pressure | | | | | | | | |
| *Elevated* | 15.4% | 39 | 8.3% | 12 | | 20.0% | 5 | 0.77 |
| *Systolic hypertension* | 33.3% | 39 | 16.7% | 12 | | 33.3% | 6 | 0.53 |
| *Systolic blood pressure percentile* | 71 | 38 | 65 | 12 | 0.5637 | 76 | 5 | 0.7247 |
| Diastolic blood pressure | | | | | | | | |
| *Elevated* | 2.6% | 39 | 25.0% | 12 | | 0.0% | 5 | 0.025 |
| *Diastolic hypertension* | 25.6% | 39 | 25.0% | 12 | | 50.0% | 6 | 0.45 |
| *Diastolic blood pressure percentile* | 72 | 38 | 83 | 12 | 0.1532 | 75 | 5 | 0.7714 |
| **Biochemical measurements in serum** | | | | | | | | |
| T3 | | | | | | | | |
| *Absolute (nmol/L)* | 4.61 (4.05–5.48) | 92 | 4.64 (4.01–5.34) | 20 | 0.011 | 3.72 (3.19–4.17) | 15 | 0.0007 |
| *Elevated* | 92.6% | 94 | 95.0% | 20 | | 93.3% | 15 | 0.93 |
| FT4 | | | | | | | | |
| *Absolute (pmol/L)* | 8.7 (2.1) | 132 | 9.3 (2.3) | 30 | 0.43 | 10.2 (2.1) | 20 | 0.011 |
| *Reduced* | 91.2% | 137 | 90.3% | 31 | | 63.6% | 22 | 0.0010 |
| rT3 (nmol/L) | 0.10 (0.07) | 62 | 0.12 (0.08) | 17 | 0.54 | 0.18 (0.09) | 10 | 0.0047 |
| TSH (mIU/L) | 3.3 (2.1–4.4) | 132 | 2.7 (2.1–4.5) | 30 | 0.30 | 1.6 (1.2–3.9) | 19 | 0.042 |
| SHBG | | | | | | | | |
| *Absolute (nmol/L)* | 257.7 (88.3) | 61 | 181.5 (90.7) | 20 | 0.0035 | 160.9 (83.4) | 6 | 0.033 |
| *Elevated* | 91.8% | 61 | 80.0% | 20 | | 66.7% | 6 | 0.11 |
| Creatinine (x LLN) | 1.32 (0.41) | 56 | 1.23 (0.47) | 16 | 0.75 | 1.20 (0.47) | 6 | 0.99 |
| CK | 132 (115) | 64 | 85 (33) | 16 | 0.23 | 69 (32) | 6 | 0.31 |
| Total cholesterol (x LLN) | 1.32 (0.33) | 46 | 1.30 (0.35) | 14 | 0.97 | 1.33 (0.38) | 5 | 1.00 |

Data are median (interquartile range[IQR]), n (%), or mean (SD). Loss-of-function classification is based on residual T3 uptake capacity in COS-1 cells. Please note that most parameters have not been captured in all patients.

[a]Cardiac arrhythm includes supraventricular tachycardia, atrial fibrillation, (non-sustained) ventricular tachycardia, ventricular bigeminy. Abnormal conduction is defined as AV-block, (incomplete) bundle branch block. Prolonged QTc-interval is defined according to age-specific cut-offs. Please note that patients may be treated with medication affecting QTc-interval.

For proportional data, Chi-square tests were used and p values in the last column indicate the presence of statistically significant differences between the different LoF categories. In all other cases, One-way ANOVA (or its non-parametric alternative) were used with Tukey's post-tests; indicated are the p-values for comparisons of severe *versus* moderate and severe *versus* mild LoF categories. Source data are provided as a Source Data file.

inclusive collaborative effort to obtain comprehensive and systemic information on nearly all known patients worldwide together with extensive functional and computational analyses, offers a generalizable approach for other rare disorders, thereby translating rare disease policy calls into practice[20–22].

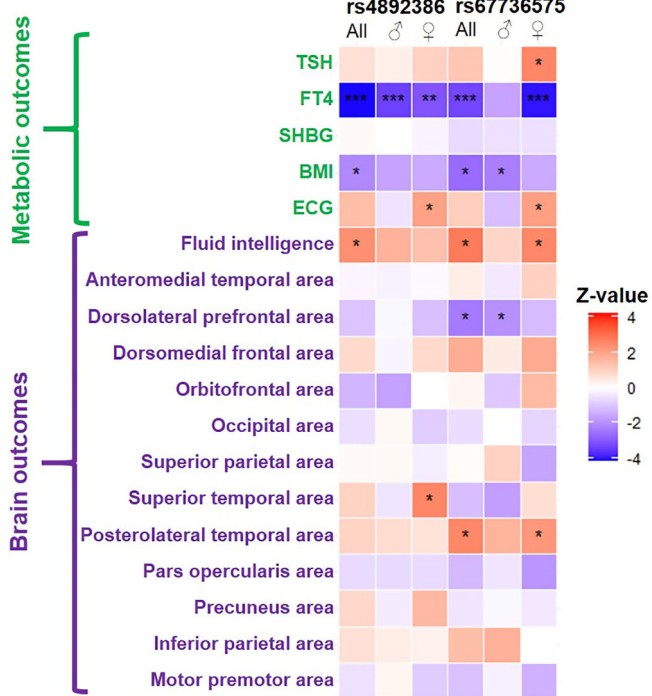

**Fig. 3 | Association of SNPs in _SLC16A2_ and relevant traits across the UKBiobank.** Association of rs4892386 and rs67736575 and different traits in participants of the UK Biobank was assessed using a multiple linear regression model with additive genetif effects, in sex-specific and joint –analyses. Multiple-testing adjustment was applied at 3 levels, nominal, with $P < 0.05$, denoted by *; categorical, denoted by **, for metabolic/thyrotoxic traits at $P < 0.005$ (5 traits), and for brain outcomes at $P < 0.0019$ (13 traits), and study-wise, with $P < 0.0014$ denoted by *** (18 traits). Exact P values are provided in Supplementary Table 5.

The functional impact of MCT8 variants was directly linked to the severity of many disease features. The LoF classification (mild, moderate, severe) predicted survival and key neurodevelopmental (e.g. motor skills and seizures), anthropometric (e.g. bodyweight, head circumference), thyrotoxic (e.g. premature atrial complexes and SHBG concentrations) features and thyroid hormone concentrations. Our observations may personalize clinical management, for example, by intensifying nutritional management in patients with a severe LoF mutation as body weight is a strong predictor of survival in these patients[14]. Also, a lower threshold to search for epileptic seizures and cardiovascular abnormalities may be indicated in patients with a severe LoF. Furthermore, it may improve a personalized counselling of parents on the disease course and inform relevant outcome measures for future clinical trials. Our genotype-phenotype correlations were corroborated by observations in non-diseased individuals in whom common genetic variation in MCT8 mildly phenocopied some disease features in patients. This illustrates how observations in a rare disease can contribute to understanding a gene's function and phenotypic consequences at the population level.

We also showed equal effectiveness of the MCT8-independent T3 analogue Triac treatment on different thyrotoxic features across different LoF classifications. As positive effects of Triac in animal models have only been shown for fully inactivated Mct8 (severe LoF)[23,24], our real-world data in humans infer that Triac treatment can be applied irrespective of the underlying mutation to ameliorate the thyrotoxic phenotype.

The integration of functional studies, Ala-scanning data and homology modeling allowed us to delineate the molecular mechanisms underlying pathogenic variants. Many pathogenic missense variants occurred at critical residues required for normal function as Ala replacement also resulted in abnormal thyroid hormone transport. For example, the replacement of D498 by Asn (patient-derived mutation) or Ala both abolished thyroid hormone transport, while substitution by Glu (having the same charge as Asp) did not affect function. The vast majority of pathogenic missense variants at non-critical residues had decreased plasma membrane expression, indicating that the substituent disrupted protein stability or subcellular trafficking. Our structure-function mutant library identified the properties of critical residues that are relevant for MCT8 function and could be employed to inform the development of tailored therapies (e.g. to selectively target candidate mutants for chaperones mutant-stabilizing properties)[25].

**Table 3 | Triac treatment – Loss-of-function correlations**

| | Severe loss-of-function $N = 59$ | | Moderate loss-of-function $N = 15$ | | | Mild loss-of-function $N = 6$ | | |
|---|---|---|---|---|---|---|---|---|
| | | _N_ | | _N_ | _p_ value | | _N_ | _p_ value |
| **Effect of Triac treatment** | | | | | | | | |
| Triac dose (ug/kg/d) | 40.5 (31.1–64.2) | 59 | 41.0 (28.5–54.3) | 15 | | 19.1 (14.0–48.3) | 6 | 0.076 |
| Treatment duration (mo) | 21.9 (12.8–34.3) | 59 | 18.8 (12.0–52.0) | 15 | | 50.6 (20.8–73.3) | 6 | 0.20 |
| Δ serum T3 (nmol/L) | −3.07 (−1.41) | 59 | −2.87 (−1.12) | 15 | 0.87 | −1.91 (−1.12) | 6 | 0.12 |
| Δ serum FT4 (pmol/L) | −6.2 (−2.5) | 59 | −6.3 (−2.2) | 15 | 0.98 | −5.1 (−1.5) | 6 | 0.56 |
| Δ serum TSH (mIU/L) | −2.13 (−1.94) | 57 | −2.38 (−1.52) | 15 | 0.88 | −1.10 (−0.97) | 6 | 0.39 |
| Δ serum SHBG (nmol/L) | −33.7 (−75.8) | 43 | −42.7 (−61.1) | 13 | 0.91 | −19.0 (−41.0) | 6 | 0.88 |
| Δ body weight-for-age[a] | 0.24 (1.68) | 52 | 0.15 (1.37) | 13 | 0.98 | −0.15 (1.15) | 6 | 0.84 |
| Δ body weight-for-age compared to natural history[b] | 0.76 (1.76) | 50 | 0.57 (1.49) | 12 | 0.93 | 0.33 (0.57) | 4 | 0.87 |
| Δ heart rate-for-age | −0.50 (−1.42) | 43 | −0.28 (−1.95) | 12 | 0.90 | −1.27 (−1.40) | 5 | 0.54 |

Data are median (interquartile range[IQR], n (%), or mean (SD). Loss-of-function classification is based on residual T3 uptake capacity in COS-1 cells. Please note that most parameters have not been captured in all patients.

[a]Patients treated shorter than 6 months were excluded from analyses.

[b]Patients older than 18 years were excluded as no natural history data is available.

One-way ANOVA (or its non-parametric alternative) were used with Tukey's post-tests in case significant differences between LoF categories were present; indicated are the p-values for comparisons of severe _versus_ moderate and severe _versus_ mild LoF categories. Source data are provided as a Source Data file.

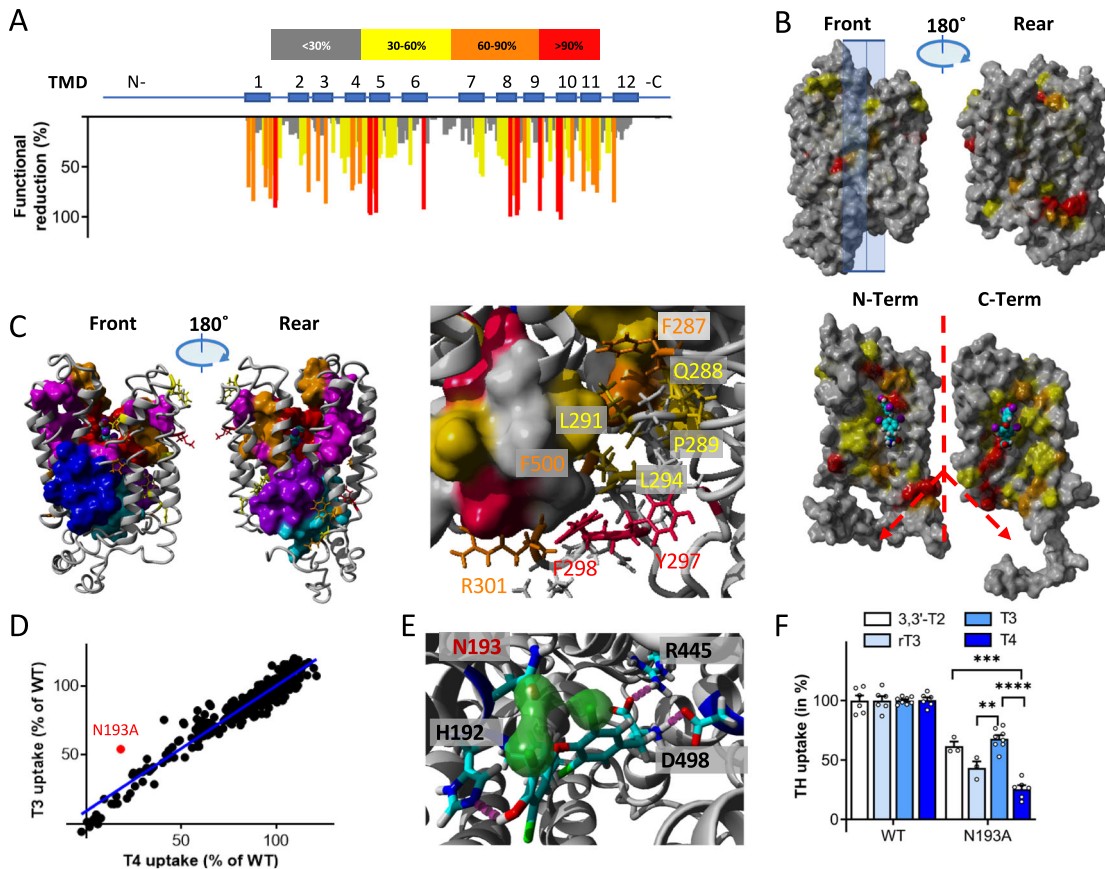

**Fig. 4 | Structural and functional features of Ala variants. A** Functional impact of MCT8 Ala variants on T4 transport in JEG-3 cells, shown as functional reduction compared to WT (red: >90–100%, severe impact; orange: 60–90%, moderate impact; yellow: 30–60%, mild impact; grey: 0–30% impact with wild-type MCT8 set as 0% impact). Grey box indicates boundaries for Ala scanning (Pro169–His575). **B** Color-coded mapping (see **A**) of functional impact identified through alanine scanning onto the MCT8 homology structure. (upper panel) frontal view and rear view of MCT8 and (lower panel) inside views (vertical section of frontal view) of the N-terminal (left) and C-terminal (right) halves. **C** Critical functional domains in MCT8: 1) residues at substrate binding center (red), 2) channel-facing residues out substrate binding center (orange), 3) residues supporting substrate-interacting residues in group 1) and 2) (magenta), 4) cluster 1 composed of TMD5 and TMD 8 (purple), 5) cluster 2 composed of TMD2 and TMD11 (dark blue), and 6) a linker region connecting clusters 1 and 2 (light blue), as well as 7) residual residues (side-chains indicated as sticks) and zoom-in of the linker region (group 6) composed of TMD4 (and partly by TMD10) connecting clusters 1 and 2, with residues color-coded according to (B). See

Supplemental Figs. 21–25 for representation of all critical domains. **D** T4 *vs* T3 transport capacity in Ala variants indicates discordant transport for N193A. **E** Halogen-bonds between Asn193 and iodothyronines. Prior to arriving at the substrate binding center, the C5-iodine of T4 and the side-chain oxygen of Asn193 are in close structural proximity (~ 3.1 Å) with overlapping Van der Waals radii and an σ-hole angle of 130–150°, which is optimal for the formation of a halogen bond between an iodobenzene and the side-chain oxygen of Asn (Ref: [40]). Simultaneously, the σ-hole of the C5'-iodine is perpendicular to the side-chain nitrogen of Asn193 at a distance of 3.3 Å, allowing the formation of a second halogen-bond that directing the large outer ring of T4. **F** Transport of iodothyronines with (rT3 and T4) and without (3,3'-T2 and T3) saturated outer ring in COS-1 cells expressing WT or N193A MCT8 and CRYM. For all substrates, values are expressed relative to WT uptake levels (set at 100%). Data were derived from ≥3 experiments with technical duplicates. *P* values were calculated using one-way ANOVA with Tukey's post-test, ** *p* < 0.005 *** *p* < 0.001 and **** *p* < 0.0001. Exact *P* values are provided in Supplementary Table 9. Source data are provided as a Source Data file.

Interrogation of functional data and structural modeling allowed us to identify critical regions of the MCT8 protein. Apart from residues making up, supporting or flanking the substrate binding center, we identified two critical clusters (TMD5/8 and TMD2/11) within the proposed rocker-helices connected by a linker (second half of TMD4/ICL2). We speculate that this linker is crucial in transducing substrate binding to conformational changes in the transporter. Of note, these regions contain the patient-derived mutational hotspots. Despite mutations in several members of the MCT family being associated with clinical disease[26,27], no large-scale mutational studies have been undertaken on any of the MCTs, and as such little is known about functionally important regions within MCT proteins. Given the high conservation among members of the MCT family, our structure-function data may serve as a resource for other MCTs beyond MCT8.

Different strategies are currently explored to address the pressing challenge of correct interpretation of genetic variation in this era of

massive genome sequencing. Deep mutational scans allow the simultaneous functional characterization of thousands of missense variants of a protein, but are dependent on the availability of relevant assays and are laborious in nature. Computational methods have recently been boosted by leveraging unsupervised deep generative models based on evolutionary data[19]. Here, to overcome the limitations of the individual experimental and computational approaches, we combined functional data with different machine learning models, thereby establishing a disease-specific optimized variant severity classifier. The addition of other features to our unsupervised model substantially improved our classifier, implying that generic and protein-wide variant effect predictors can still benefit from the addition of protein-specific features. Moreover, although disease-specific variant pathogenicity classifiers have been generated for rare diseases before, the generation of a disease-specific dual pathogenicity-severity classifier is exceptional, and may prompt the generation of such classifiers in other

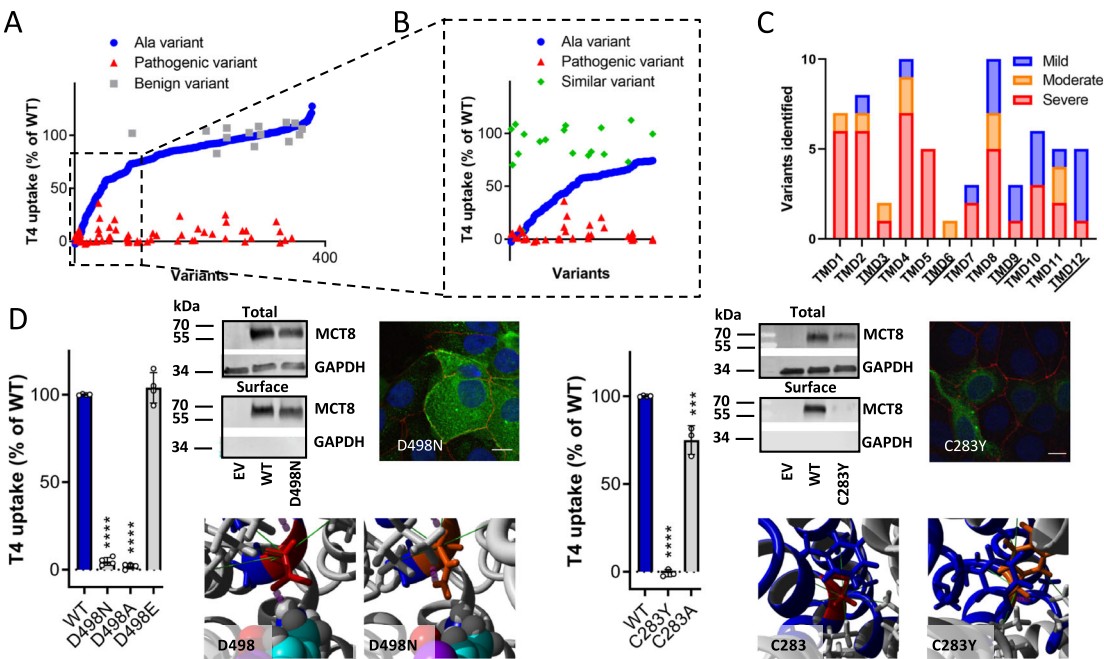

**Fig. 5 | Molecular characterization of patient mutants. A** T4 uptake capacity of all tested Ala variants (blue; ranked from 0% to 100% transport capacity), LoF patient variants (red), and benign non-synonymous missense variants (grey). Patients variants affecting residues within the grey box are likely pathogenic due to the loss of a critical native residue; patient variants affecting other residue are likely pathogenic due to the introduction of an unfavorable residue. **B** T4 uptake capacity in selected panel (dashed box in Fig. 5**A**) of Ala variants (blue), LoF patient variants (red) and artificial variants, where the native residue at the position of patient variants was replaced by a residue with similar properties (dark grey). **C** distribution of pathogenic variants identified in patients with MCT8 deficiency among the 12 transmembrane domains (TMDs) of MCT8. Variants were categorized based on residual T3 uptake in COS-1 cells with <20%, severe LoF (red); 20–40%, moderate LoF (orange); 40–75%, mild LoF (blue). **D** Exemplary mutations for different pathogenic mechanisms. (left panel) Like the patient variant D498N with normal membrane expression, there is complete LoF in artificial variant D498A but preserved transport capacity in artificial D498E which has similar properties,

highlighting the critical role of the original residue. (right panel) In contrast to the complete LoF patient variant C283Y which has low membrane expression, transport capacity is largely preserved in the artificial variant C283A, highlighting the damaging effect of the substituent rather the loss of the native residue. T4 transport capacity (mean±s.e.m.) in JEG3 cells expressing WT (set as 100%) or mutant MCT8. Data were derived from ≥3 experiments with technical duplicates. P values were calculated using one-way ANOVA followed by Tukey's multiple comparisons test; * $p < 0.05$, *** $p < 0.001$. Exact P values are provided in Supplementary Table 9. Source data are provided as a Source Data file. Immunoblot of total lysates and cell surface biotinylated fraction of COS-1 cells expressing WT or mutant MCT8. GAPDH was used as a loading and purity control. Immunocytochemistry in JEG-3 cells showing co-localization of MCT8 (green) and the membrane marker ZO-1 (red) for WT and mutant MCT8. Nuclei were stained with DAPI (blue). Scale bar corresponds to 15 uM. MCT8 homology model highlighting the affected residues (red) and the impact of the LoF patient variants (orange). Full data on all LoF patient variants is available in Supplemental Fig. 5.

(rare) diseases. Severely pathogenic variants are segregated with >90% accuracy by our dual classifier, greatly outperforming other computational approaches, particularly in pivotal regions of the protein.

We acknowledge several limitations. First, clinical data was not acquired in a uniform way for a substantial part of the patients. Second, we did not incorporate data on females with mutations in MCT8. Female carriers of MCT8 mutations with unfavorable X-inactivation may present with varying clinical features[28–30]. The degree of unfavorable X-inactivation as well as the tissue-specificity differs across such patients. Therefore, we decided not to include such data in the present report as this would hamper proper comparison with males with MCT8 deficiency. Third, we acknowledge different types of potential confounders (e.g. socio-economic status and health care disparities). Other not yet elucidated disease-modifiers may account for variation within LoF groups. Fourth, the clinical application of our pathogenicity-severity classifier cannot reach its full potential unless early diagnostics is implemented. Potential strategies might include neonatal screening[31] or prenatal testing, although the utilization of thyroid function tests to identify female carriers, with the aim of initiating potential antenatal therapies in affected fetuses has not been found to be feasible due to the lack of significant differences compared to non-carriers. Fifth, the accuracy of the severity module in the classifier could improve with the identification of new disease-causing variants, particularly those with mild and moderate LoF.

Our integrative multi-disciplinary and worldwide collaborative approach combining deep phenotyping data with a series of functional and computational tests and with outcomes in multiple population cohorts, enabled us to: (i) understand the divergent clinical phenotypes of MCT8 deficiency, which may assist with personalization of clinical management (e.g. anticipatory guidance and counseling) and guidance of future clinical trials; (ii) assess therapy effectiveness across different LoF categories; (iii) advance structural insights of MCT8, to facilitate the development of tailored therapies; (iv) create a high-quality disease variant and severity classifier, informing variant interpretation in the diagnostic trajectory; (v) leverage information on the role of MCT8 in non-affected individuals in the population. Our information-dense mapping provides a generalizable approach to advance multiple dimensions of rare genetic disorders.

## Methods
### Ethical considerations
This study conforms to the Declaration of Helsinki, Good Clinical Practice guidelines. Some centers required and obtained approval by Ethics Committees/Institutional Review Boards. Some other centers required and obtained a waiver from approval. In most centers, however, no ethical approval or the provision of a waiver was required according to their (national) regulations, since this study used anonymized patient data retrieved in the context of clinical care. Informed

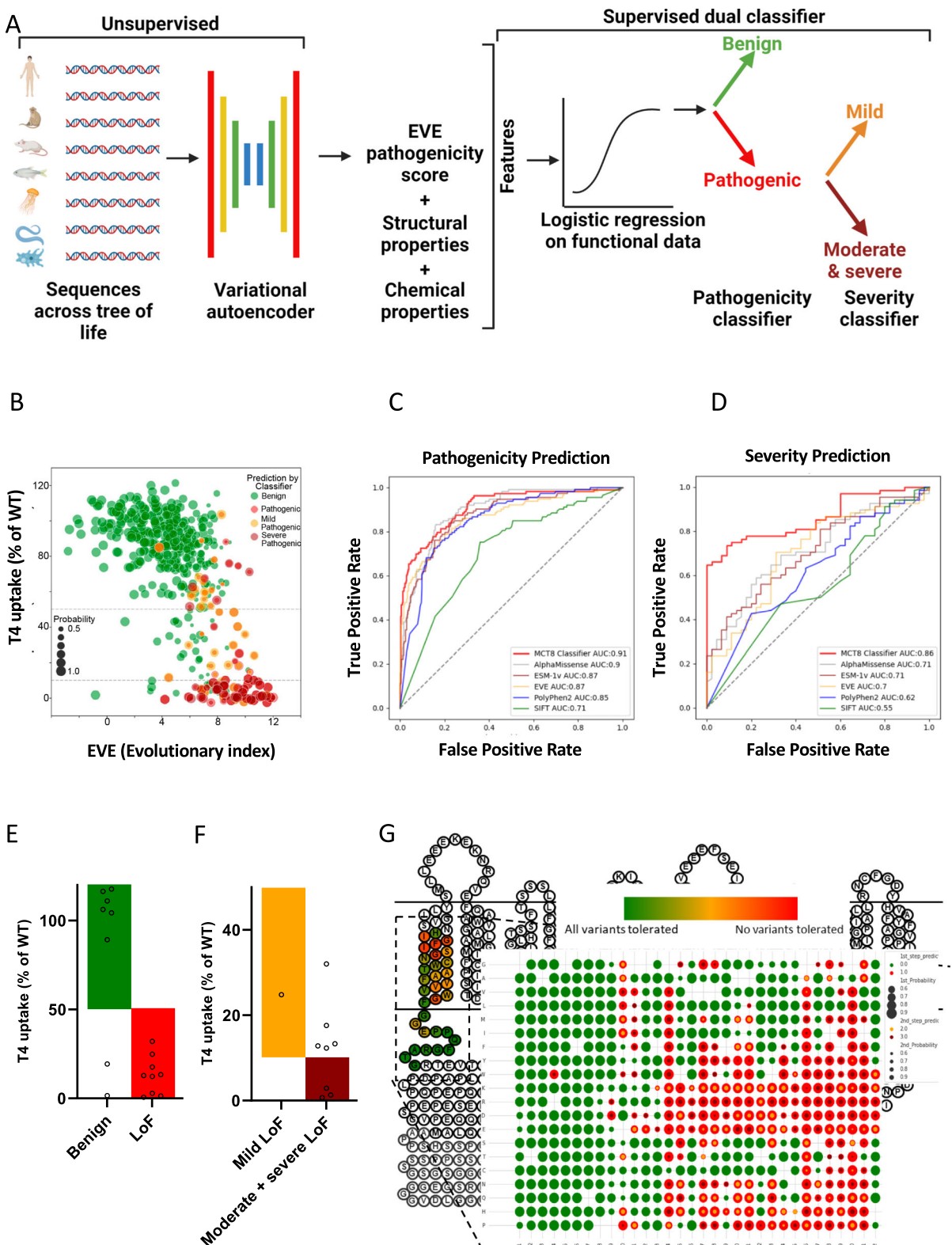

consent was obtained from the parents or legal representatives of all enrolled patients, unless the relevant institutional review board or local regulators had authorised the use of anonymised patient data without additional consent[14,15,17]. Data on Triac effects were obtained from previous studies[15,17].

Ethics Committees/Institutional Review Boards that provided approval are the Institutional Review Boards of Louisville, USA; MultiCare IRB, Tacoma, USA; Oregon Health & Sciences University,

Portland, USA; Ethics Committees from Children's Health Queensland Human Research Ethics Committee, South Brisbane Queensland, Australia; the Bioethics Committee for Scientific Research Medical University of Gdansk, Poland; the University Children's Hospital Zurich, Switzerland; Hunter New England Human Research Ethics Committee, Australia; Comitato Etico Milano Area 1 (2018/ST/117), Milan, Italy; and in the context of a previous clinical trial the Ethics Committees from Erasmus Medical Center, The Netherlands; Panorama

**Fig. 6 | MCT8 variant classifier for benign *vs* pathogenic and for different LoF classes. A** Design of the dual pathogenicity-severity MCT8 variant classifier. Pathogenicity of all variants is predicted by the first algorithm (MCT8 pathogenicity classifier); variants that are predicted pathogenic are segregated by the second algorithm in having mild or combined moderate and severe LoF, thereby assessing disease severity (MCT8 severity classifier). Created in BioRender. Visser, E. (2023) BioRender.com/s06h168. **B** Pathogenicity and severity prediction of all functionally evaluated variants by the unsupervised approached based on EVE; higher number denotes a stronger evolutionary constraint. Color and size of dots correspond to prediction by the dual pathogenicity-severity classifier. Dashed lines represent cut-off values for different LoF classes. **C** Performance of the MCT8 pathogenicity classifier for all functionally evaluated variants as shown by receiver operating characteristic (ROC) curve, with direct comparison to the unsupervised machine learning tool (EVE) and commonly used pathogenicity prediction tools. Validation of the classifier was done with a 10-fold cross-validation. AUPRC curves are presented in Supplementary Fig. 28a. **D** Performance of the MCT8 severity classifier for all functionally evaluated variants (that were predicted pathogenic in **C** as shown by receiver operating characteristic (ROC) curve, with direct comparison to EVE. Validation of the classifier was done with a 10-fold cross-validation. AUPRC curves are presented in Supplementary Fig. 28b. **E** Validation of the MCT8 pathogenicity classifier by functional evaluation of 11 novel patient variants and 6 random artificial variants. Green: wild-type function, red:pathogenic LoF (based on residual uptake capacity). **F** Validation of the MCT8 severity classifier by functional evaluation of the variants predicted pathogenic in **E**. Orange: mild LoF, dark red: moderate or severe LoF. **G** Predicted mutational landscape for MCT8. Example of predicted pathogenicity and severity of all variants on one transmembrane domain (region G161 to H192). The background shows a 2D structural model of MCT8, with colors indicating overall tolerability of missense variants in the amino acid residues in this region (red: no variants tolerated; green: all variants tolerated). The front shows the mutational landscape of amino acid residues in this domain (green: predicted benign; red: predicted pathogenic; yellow in red: predicted mild and moderate; brown in red: predicted severe; the size of the circle denotes the probability of correct prediction: larger circle, higher probability). Source data are provided as a Source Data file.

---

Medical Centre, Republic of South-Africa; Fakultní nemocnice v Motole, CZ Czech Republic; Alexandy Obregia Clinical Psychiatric Hospital, Romania; AZ Sint-Jan; UZ Brussel, Belgium; Necker University Hospital Paris, France; Toulouse University Hospital, France; Charite-Universitätsmedizin Berlin, Germany; U.O. Neuropsichiatria Infantile – Instituto Neurologica Carlo Besta, Italia; Ospedale Pediatrico Bambino Gesù, Italia; Addenbrooke's Hospital, Cambridge, United Kingdom.

Ethics Committees/Institutional Review Boards that provided a waiver from approval or consent are the Institutional Review Board of Children's Hospital of Philadelphia, USA (IRB #17-014224) the Ethics Committees at Erasmus Medical Center, The Netherlands (MEC-2015-362); SRCC Children's Hospital, India; St. John's Medical College, Bengaluru, India.

In all other centers, this was not required.

### International consortium on MCT8 deficiency

This international study was initiated on 14 October 2014 by establishing a consortium of centres where patients with MCT8 deficiency were managed[14]. The key inclusion criterion was genetically confirmed MCT8 deficiency. Additionally, data for first-degree and second-degree male relatives with clinical MCT8 deficiency (when genetic testing was not available at that time) were included. There were no exclusion criteria. Our cohort consisted of patients who had been enrolled in the Triac Trial I[15], patients who participated in the named patient program for Triac treatment[17], and historical cases for whom the Erasmus University Medical Center (Erasmus MC; Rotterdam, Netherlands) fulfilled a consultancy role following the first reports of MCT8 deficiency in 2004[11,12]. The group of historical cases therefore contained patients who were alive and patients who were already deceased at time of enrolment. A subgroup of participants has been reported before with available individual case descriptions (*n* = 48), or on an aggregated level (*n* = 107). For these patients, updated and exhaustive data were collected.

### Study objectives

Our first objective was to analyse the relation between variants in *SLC16A2* and disease severity. To this purpose we determined the functional impact of variants identified in MCT8 patients using well-established cell-based disease models and assessed their relationship to overall survival (primary outcome) as well as 32 other core clinical and biochemical disease features (secondary outcomes; defined below). Further, we evaluated the association of common genetic variation in the *SLC16A2* locus in healthy individuals with serum thyroid function tests and various functional and imaging brain outcomes.

The second objective was to determine the effects of Triac on peripheral phenotypic features differed among patients with different LoF classifications.

The third objective was to identify critical residues in the MCT8 protein and delineate pathogenic mechanisms of variants identified in patients with MCT8 deficiency.

The fourth objective was to establish a dual pathogenicity-severity classifier that allows accurate prediction of functional impact of any missense variant in the MCT8 protein.

### Study procedures and outcomes

An overview of clinical study assessments and investigations is provided in Supplementary Tables 1–3. Clinical and biochemical features were acquired in patients who did not receive potential disease-modifying interventions such as treatment with thyroid hormone analogues. The acquisition and documentation of disease features in our international consortium has been described in detail before[14,15]. These data were captured according to standard operating procedures at the baseline visit for the Triac Trial [NCT02060474][15] or in named-patient Triac treatment programs[17], and did not receive any potential disease-modifying treatments (e.g. (anti-)thyroid drugs or thyroid hormone analogues) at time of evaluation. To reduce selection bias and taking advantage of previously reported data of additional patients, we extended this real-time cohort by adding data from previously documented cases captured from a systematic literature review (see below and Supplementary Fig. 1). These meta-analysed data on disease outcomes were used in the correlation analyses with the functional impact of variants at MCT8 protein level.

For all cases included, we aimed to retrieve demographic characteristics and genetic, clinical, neurodevelopmental, neuroimaging and biochemical features (see Supplementary Tables 2, 3), as well as data on survival, defined as age at date last known alive. The following characteristics were used in the genotype-phenotype analyses. The following data relating to neurocognitive parameters were collected: neurological clinical examination (i.e. presence of hypotonia, primitive reflexes, muscle hypoplasia, dystonia, spasticity, plantar extension response, microcephaly), MRI data (i.e. delayed myelination), functional data (i.e. EEG-proven seizures). Neurodevelopmental outcome measures included the acquisition of developmental milestones (i.e. head control, speech, independent sitting, independent walking), and scores on well-defined developmental assessments (i.e. Gross Motor Function Measure [GMFM]-88[32], Vineland Adaptive Behavior scales [VABs]-II[33] and Bayley Scales of Infant Development [BSID]-III[34]). For genotype-phenotype correlation analyses on developmental outcomes, patients who were not anticipated to have reached the outcomes at the age of assessment (based on development of healthy individuals) were excluded. The following data relating to metabolic and thyrotoxic features of MCT8 deficiency were collected: the presence of underweight (weight-for-age z-score <-2SD; overall, and specifically in early childhood, defined as 1–3 years of age), short stature

(height-for-age z-score <−2SD), feeding problems (parent-reported outcome), tube feeding, gastro-oesophageal reflux disease, and elevated systolic or diastolic blood pressure; as well as results of 24 h cardiac telemetry and resting electrocardiogram (i.e. heart rate, presence of tachycardia, cardiac arrhythmia's, conduction abnormalities, or prolonged QTc-interval). When appropriate, Z scores were calculated to enable comparison to normal development. Body weight-for-age Z scores were calculated using TNO growth calculator and heart rate-for-age Z scores were calculated using the Boston Z score calculator as used previously[14,17]. Weight-for-age and height-for-age Z scores were compared to the available natural history data obtained in our MCT8 deficiency cohort (Triac-naive state)[14]. The difference to the natural history curve was determined for each subject and used for comparison, as described before[17]. Given the scarcity of natural history data in subjects aged above 20 years (upper limit of the applied Z score calculators), subjects older than 20 years were set at 20 years of age. Elevated systolic and diastolic blood pressure were defined using the guidelines from the American Academy of Pediatrics[35] and the American College of Cardiology and American Heart Association[36]. Tachycardia was defined as a resting heart rate above the 90th percentile for the corresponding age, with cut-offs as previously described[37]. Biochemical analyses included endocrine parameters (i.e. T3, FT4, rT3, and TSH) and serum markers that reflect thyroid hormone action in peripheral organs (i.e. SHBG, creatinine, CK, and total cholesterol). For patients enrolled in the Triac Trial, the Triac named patient program, or for whom Erasmus MC fulfilled a consultancy role, all biochemical analyses, including thyroid function tests, had been undertaken in a central laboratory (Erasmus MC, Rotterdam, The Netherlands), to minimize the effects of inter-assay variation, which is particularly present among the various antibody-based methods being used for measuring thyroid hormones. When evaluating the effect of Triac treatment on serum T3 concentrations, we used a previously documented algorithm to correct for interference of Triac in the T3 assay[15].

Data from patients followed up between January 2003 and December 2020 were retrieved from October 2014 to January 2021. To minimize the confounding effect of missing data in medical records of each patient, the presence or absence of a specific parameter was only documented when it was specifically recorded as being absent or present in the medical file.

### Procedures in systematic literature review
**Search strategy.** A computerized search of public databases Embase, PubMed, Google Scholar, and Scopus from January 2004 to August 2020 was conducted. The search terms were as follows: MCT8, SLC16A2, AHDS, Allan-Herndon-Dudley syndrome, and MCT8 deficiency. Full-text articles of abstracts were then selected, retrieved, and assessed for eligibility considering the established criteria detailed above. Inclusion was based on final consensus between at least two authors. The reference lists of all articles selected for review, and the full texts of the potentially relevant studies were also examined.

**Inclusion criteria.** Studies were included if they: 1) reported the SLC16A2 variant at either cDNA or protein level, 2) contained clinical descriptions of original cases, and 3) were peer-reviewed or published as an abstract in a conference paper.

**Data extraction.** All relevant data were extracted from selected articles or abstracts and imported into a central database. Data were cleaned and cross-checked to ensure that no individual was recorded more than once. The following information was used to identify potential duplicates: mutation (cDNA or protein change), age at assessment (in combination with publication date), inheritance information, as well as the author list. Once a potential duplicate was identified, the most recent information for that individual was included in the review and all previously reported information was integrated and cross-checked.

The study procedures of a clinical trial and a real-world retrospective cohort study, which both documented the efficacy of Triac in patients with MCT8 deficiency, were described before[15,17]. Briefly, through combination of the studies, we utilized data from Triac-treated patients with MCT8 deficiency with available data ($n = 85$; without splice site mutations: $n = 80$) in 33 sites. For the current study, we analyzed the changes in serum T3, TSH and fT4 concentrations, bodyweight-for-age, heart rate and serum sex hormone binding globulin (SHBG) concentrations, all from baseline to last available measurement, upon stratification according to severe, moderate or mild LoF classification. These parameters were selected based on their clinical relevance and availability of sufficient observations across different LoF classifications.

**Selection of SLC16A2 variants in unaffected populations.** We screened for non-synonymous variant data from control-only individuals in gnomAD, a database that aggregates sequencing studies[38]. Because individuals with pediatric disorders are excluded from gnomAD, and because these individuals were specifically accrued as unaffected controls, they were assumed to be free of MCT8 deficiency. Variants that occurred in at least two male individuals and had an allele count >10 were included.

**Meta-analysis and genotype-phenotype analyses.** We utilized data from patients with MCT8 deficiency through our international consortium consisting of 53 sites in 23 countries. To this cohort we added patients identified through a systematic literature review (up to 01-01-2020), for whom data on clinical outcomes were extracted from the original publication. Data accrued from the international consortium on MCT8 deficiency and from the literature review were assembled into one database using a uniform coding for all parameters.

All disease-causing and benign MCT8 variants were functionally tested, except for variants that lead to a premature truncation before transmembrane domain (TMD) 12 as such truncations have been previously shown to be fully inactive[39,40]; hence, these mutations were regarded as having 0% residual thyroid hormone transport. Patients with splice site variants were excluded from clinical correlation analyses, as the impact of such variants on splicing efficacy can be heterogeneous across tissues and therefore difficult to predict in cell-based models.

Our objective was to correlate residual transport activity of pathogenic MCT8 variants in transfected COS-1 (primary model) and JEG-3 cells (confirmatory model), and patient-derived fibroblasts with disease features. Patients were stratified across different LoF classifications: severe LoF ( < 20% residual T3/T4 transport as determined in transfected COS-1 cells and <5% residual T3/T4 transport in transfected JEG-3 cells), moderate LoF (20–40% residual T3/T4 transport in transfected COS-1 cells and 5–10% residual T3 transport in transfected JEG-3 cells) and mild LoF (40–75% (T3) or 40–65% (T4) residual transport in transfected COS-1 cells and 10–50% (T3) or 10–40% (T4) residual transport in transfected JEG-3 cells). The LoF classifications were chosen to enable relevant discrimination across groups, guided by limited small-scale functional and clinical interrogations[41,42]. Due to the limited number of patient fibroblasts ($n = 34$), severe LoF was defined as <50% and mild LoF as 50–75%. We correlated T3 transport with survival and the following disease characteristics: a. neurodevelopmental scores on the subdomains of the Bayley Scales of Infant Development (BSID)-III[34], Gross Motor Function Measure (GMFM)-88 scale[32] and Vineland Adaptive Behaviour Scales (VABS)-II[33] as well as clinical features (attainment of head control, sit, walk or speech); b. neurological features (hypotonia, primitive reflexes, muscle hypoplasia, dystonia, spasticity, plantar extension response, seizures, delayed myelination, microcephaly); c. thyrotoxic signs (underweight, short stature, feeding status, heart rate, premature atrial complexes, relevant cardiac abnormalities (arrhythmia, abnormal conduction, or

prolonged QTc), systolic blood pressure, diastolic blood pressure; d. biochemical measurements (T3, FT4, rT3, TSH, SHBG, CK, total cholesterol); e. treatment effects of Triac on T3, FT4, TSH, SHBG, body weight and heart rate.

**Statistical procedures.** We summarised continuous variables as mean (SD) or median (range). Survival was defined as the age at the last date known alive. In analyses of developmental outcomes, patients who were not anticipated to have reached the outcomes on the age of assessment (based on development of healthy individuals) were excluded. Residual transport capacity was set to 0% for large deletions and premature frameshifts (protein termination before residue 575 has been shown deleterious previously)[40]. Patients with splice site variants were excluded from clinical correlations.

The degree of thyroid hormone (T3 and T4) transport activity of MCT8 variants in transfected COS-1 and JEG-3 cells or patient-derived fibroblasts was correlated with survival and 32 disease features upon stratification across abovementioned LoF classifications. We assessed overall survival and compared patients with mutations across different LoF classifications using log-rank analysis. Clinical outcomes were compared between three LoF classifications (transfected cells-based stratification) using Chi-square tests (proportional data) or one-way ANOVA with Tukey's posttests (continuous data), and between two LoF classifications (fibroblast-based stratification) using Students T-tests; or their non-parametric alternatives. Descriptive statistics and scatterplots were generated for variables used in the analyses to ensure that the data met the criteria for the use of parametric tests. In case the normality assumption was not met, non-parametric tests were utilized. All analyses were performed using Graphpad Prism (version 9.0), and p values ≤ 0.05 were considered significant.

**Functional studies**

**Materials.** Nonradioactive iodothyronines, silychristin, bovine serum albumin (BSA), and D-glucose were obtained from Sigma Aldrich (Zwijndrecht, The Netherlands [NL]). [125I]-T3 and [125I]-T4 were produced as previously described (e.g. ref.[9,43]). An overview of the antibodies is provided in Supplementary Table 8. Vectashield H-1200 containing 4′,6-diamidino-2-phenylindole (DAPI) was obtained from Brunschwig (Amsterdam, NL). All cell culture flasks and plates were obtained from Greiner Bio-one (Alphen aan den Rijn, NL). X-tremeGENE9 transfection reagent was obtained from Roche Diagnostics (Woerden, NL). Sulfo-NHS-biotin was obtained from Gentaur (Eersel, NL). Neutravidin agarose was obtained from Thermo Fisher scientific (Bleiswijk, NL).

**Plasmids.** Cloning of human MCT8 in pcDNA3, and of the human cytosolic thyroid hormone-binding protein mu-crystallin (CRYM) in pSG5, has been described previously[10,43]. The generation of a wild-type (WT) human MCT8 expression construct containing a 227-nucleotide extension of the 3′UTR (further referred to as 3′UTR-MCT8) has been described before[44]. The indicated variants were introduced using QuikChange site-directed mutagenesis according to manufacturer's protocol (Stratagene, Amsterdam, NL). Primer sequences are available upon request. In line with previous studies[40], the 3′UTR-MCT8 construct was used as a template for all frameshift variants that resulted in an alternate reading frame that exceeded the natural stop codon of MCT8, whereas the regular MCT8 expression construct was used as a template for all other variants. The position of the mutations is indicated using the NM_006517.3 reference sequence and uses +1 as the A of the ATG translation initiation codon of the long MCT8 isoform, with the initiation codon as codon 1. DNA sequencing confirmed the presence of the introduced mutations and the absence of unintended mutations.

**Cell culture and transfection.** COS-1 African green monkey kidney (CVCL_0223) and JEG-3 human choriocarcinoma (CVCL_0363) cells were obtained from ECACC (Sigma-Aldrich) and were cultured and transiently transfected as previously described[44]. For thyroid hormone uptake studies, COS-1 or JEG-3 cells were cultured in 24-well plates, and transiently transfected at 70% confluence with 100 ng pcDNA3 empty vector (EV), or 100 ng WT or indicated mutant MCT8 expression construct in the presence or absence of 50 ng CRYM. We have previously excluded the presence of functional differences between the regular WT MCT8 and 3′UTR-MCT8 constructs[40]. Determination of optimal plasmid doses have been previously described[16]. For surface biotinylation studies, cells were seeded in 6-well plates (6 wells per condition, which were pooled during lysate preparation) and transiently transfected with 500 ng pcDNA3 EV, WT or indicated mutant MCT8 per well. For immunocytochemistry, JEG-3 cells were cultured in 24-well dishes on 10-mm glass coverslips coated with poly-D-lysine (Sigma-Aldrich).

Human fibroblasts were derived from skin biopsies obtained during routine clinical practice and cultured in Dulbecco's modified Eagle medium/F12 medium (Invitrogen, Breda, NL), containing 9% heat-inactivated fetal bovine serum (Invitrogen, Breda, NL), 2% penicillin/streptomycin (Roche). For uptake studies, fibroblasts were seeded in 6-well plates and grown until >95% confluence[16].

**Thyroid hormone transport studies.** Thyroid hormone transport studies were performed using well-established protocols (e.g. ref.[45]). Cells were washed once with incubation buffer (D-PBS+Ca2+/Mg2+ supplemented with 0.1% glucose and 0.1% BSA) and incubated in incubation buffer containing 1 nM (50,000 cpm) [125I]-T3 or [125I]-T4 for 30 min. Cells were then briefly washed with incubation buffer and lysed in 0.1 M sodium hydroxide. The internalized radioactivity was measured with a gamma-counter. Thyroid hormone uptake levels in human fibroblasts were corrected for total protein concentrations as measured by Bradford assay according to manufacturer's guideline (Bio-Rad, Veenendaal, NL). To determine residual MCT8 activity in human fibroblasts, T3 uptake was determined in the presence of 10 μM silychristin, a well-established MCT8-specific inhibitor[16,46].

**Surface biotinylation and immunoblotting.** Cell surface biotinylation studies were performed according to well-established protocols[16,45]. Cell surface proteins were labeled with Sulfo-NHS-biotin and lysed in IP buffer (50 mM Tris-HCl, 150 mM NaCl, 10 mM EDTA, 1% Triton X-100), containing protease inhibitor cocktail (Roche). After brief sonication, samples were clarified from nuclear debris by centrifugation (15000xg for 10 min). A 5% aliquot was used as an input control. Cell surface proteins were isolated using Neutravidin agarose beads (Thermo Fisher Scientific) and eluted in NuPAGE 1x lithium dodecyl sulfate (LDS) loading buffer (Thermo Fisher Scientific) containing 10 mM DTT by incubating the beads for 5 min at 90 °C prior to immunoblot analyses. Samples were analyzed by immunoblotting as previously described[16,47], using antibodies listed in Supplementary Table 8 and Odyssey detection methods[40].

**Immunocytochemistry.** Transiently transfected JEG-3 cells were fixed with 4% paraformaldehyde and permeabilized with 0.25% triton X-100 in PBS 48 h after transfection. Samples were blocked for 1 h at room temperature in PBS containing 2% BSA (Sigma Aldrich), and incubated overnight with rabbit anti-MCT8 (1:1000) and mouse monoclonal ZO-1 antibody (1:500), which served as a membrane marker. After secondary staining with goat anti-rabbit Alexa Fluor 488 (1:1000) and goat anti-mouse Alexa Fluor 633 (1:1000), cover slips were mounted on glass slides with Prolong Gold containing DAPI (Invitrogen) and examined as previously described[44].

**Alanine scanning.** All residues from just prior to the start of TMD1 (Pro169) until the end of TMD12 were mutated to Ala residues using standard mutagenesis procedures (e.g. ref.[47]). For the loops (Lys207-Gln216; Asp240-Thr247; Thr265-Ser269; Asp329-Thr336; Arg355-Asn386; Met417-Ile427; Pro456-Gly457; Ile478-Leu480; Glu511-Pro515; Phe547-His551) we created Ala blocks encompassing up to 5 residues. If residual transport of an Ala block was <50%, we mutated all individual residues in the block to Ala. Small loops were tested as individual variants. Functional evaluation of Ala variants was performed in JEG-3 cells using T3 and T4 as a substrate.

**Genetic look-up of the *SLC16A2* gene.** For the lookup, the X-linked *SLC16A2* gene was selected. All SNPs located within +/−100kb of the gene's transcription start and end site were included in the lookup (TSH men and women $N = 356$, TSH men $N = 339$, TSH women $N = 340$; FT4 man and women $N = 362$, FT4 men $N = 344$, FT4 women $N = 343$). Filters on minor allele frequency (MAF) > 1% and imputation quality >0.4 were applied and a false discovery rate (FDR) of 5% was used to correct for multiple testing. We selected TSH and FT4 as main traits as thyroid function tests are abnormal in virtual all patients with MCT8 deficiency. For TSH ($N = 54,288$) and FT4 ($N = 49,269$), the summary statistics of the sex-specific and joint (both sexes) GWAS meta-analyses from Teumer et al. were used[48]. For the chromosome X analyses, an additive genetic association model was calculated coding females as 0/1/2 and males 0/2, assuming full genetic dosage compensation, depending on the number of alleles on both or one chromosome X respectively[49]. Individuals with TSH values outside the cohort-specific reference range, who underwent thyroid surgery, or taking thyroid medication (ATC code H03) were excluded from all analyses.

Pre-imputation quality checks included SNPs with a call rate ≥ 95% and HWE p-value > $10^{-4}$ and minor allele frequency ≥ 0.01. Further information on quality control checks and meta-analysis of the on TSH and FT4 is described in detail elsewhere[48].

When significant results were found (FDR < 5%) for a specific trait in the sex-specific or joint analyses, additional clumping using PLINK v1.9 was performed on all SNPs in the look-up region for each analysis (i.e. males, females, both sexes) separately, assessing independently associated SNPs with UKBB as a reference (14k individuals). To include all SNPs in the clump, final significance threshold filters (p1 and p2) were set to default 1. In order to remove too correlated SNPs, no pair of SNPs survived the clump with a squared correlation above 0.01 within a window of 1 Mb. Afterwards a FDR was applied and SNPs with a FDR < 5% were considered significant. For applying Bonferroni multiple testing correction, the number of statistically independent SNPs in the specific MCT8 region was calculated using the indep option in PLINK on all SNPs in the look-up region of the specific trait. This method removes correlated SNPs (based on a multiple correlation coefficient and the (LD)). A 2000kb window shifted by 5 consecutive SNPs, and a variation inflation factor of 5 ($R^2 = 0.8$) were selected as parameters. The total count of independently associated SNPs in the look-up region was used in the Bonferroni equation (0.05/independent SNPs), as a more conservative multiple testing correction. Furthermore, SNPs were also checked if passing nominal significance ($p < 0.05$). Altogether, three SNPs were selected (rs150010878 (both sexes), rs4892386 (males only) and rs67736575 (females only)) that were significantly associated with FT4 concentrations. Given the high correlation between rs150010878 and rs4892386 (r2 = 0.98 in the European set of the 1000 genomes reference panel), follow-up analyses were only performed for rs4892386 and rs67736575.

The association of the two selected SNPs was then assessed in the UKBB, using accession IDs: 67864 and 27412)[50]. The UK Biobank (https://www.ukbiobank.ac.uk) is a multi-site cohort study consisting of 502,655 individuals aged between 40 and 69 years at baseline. The study was approved by the National Research Ethics Committee, reference 11/NW/0382, and informed consent was obtained from all participants as part of the recruitment and assessment process. Using accession ID 67864, sex hormone binding globulin (SHBG, $n = 353,029$ [46% females]), body mass index (BMI, $n = 406,975$ [46% females]), electrocardiogram heart rate during exercise (ECG, $n = 56,186$ [46% females]) and fluid intelligence score ($n = 130,134$ [46% females]) were assessed assuming an additive model, adjusting for (sex), age, research center, genotyping platform and 10 genomic principal components in individuals of European ancestry using SAIGE[51]. A similar approach was followed for brain outcomes, under accession ID 27412. Briefly, 70 different brain traits including cortical thickness, surface areas[52], sub-cortical volumes and fiber tract anisotropy variables derived from MRI ($n = ~32,000$) were tested using FastGWA on the chromosome X[53]. Assuming an additive model, joint and sex-specific analyses were adjusted for (sex), age, Euler number, 10 genomic principal components, BrainDx, mean cortical thickness and total surface area. In addition, gene-based analysis were also carried out using GCTA-fastBAT which performs a set-based enrichment analysis using GWAS summary statistics while accounting for linkage disequilibrium (LD) between SNPs of a particular region[54]. For sex-specific analyses only females or males present in this subsample were used to create sex-specific reference LD panels. Multiple testing adjustment was carried out and is reported as follows. Multiple-testing adjustment was applied at 3 levels, nominal $P < 0.05$, denoted by *; categorical denoted by **, (i.e., Metabolic [TSH, FT4, SHBG, BMI, ECG]; $P < (0.05)/(5 \text{ traits}*2 \text{ SNPs})) = P < 0.005$. Brain outcomes [fluid intelligence and surface area sizes of brain regions], $P < (0.05)/(13 \text{ traits}*2 \text{ SNPs})) = P < 0.0019$; and study-wise, denoted by ***, $P < (0.05)/(18 \text{ traits }*2 \text{ SNPs})) = P < 0.014$. Additional analyses of other explored traits are also shown on the Supplementary Fig. 11, where $P < 0.05$ is nominally significant; specific area significance (i.e., Specific brain thickness $P < (0.05)/(11 \text{ traits}*2 \text{ SNPs}) = P < 0.0023$. Subcortical volumes $P < (0.05)/(20 \text{ traits}* 2 \text{ SNPs})) = P < 0.00125$. Fiber tract anisotropy, $P < (0.05)/(27 \text{ traits}* 2 \text{ SNPs})) = P < 0.0009$). All brain MRI outcomes were adjusted for global brain sizes.

## In silico analyses

**Conservation analyses.** Conservation analyses of protein homologs were performed using the ConSurf server[55], using HMMER as homolog search algorithm, 1 iteration, an E-value cutoff of 0.0001 and UniProt used as protein database. Homologs were selected automatically by the server. All sequences closest to the reference sequence with a %ID between 35 and 100% were selected for analyses and aligned using the MAFFT-L-INS-I method. Bayesian calculation was performed using the best evolutionary substitution model. The long isoform (NP_006508.1) was used for all conservation analyses. The MCT8 subtree was extracted from the collected sequences using WASABI[56] and reanalyzed using the ConSurf server afterwards (conservation score 1). For conservation score 2, mammalian sequences were extracted manually, a multiple sequence alignment of these sequences was created using ClustalOmega[57] and analyses were performed using Bayesian calculation with the best evolutionary substitution model. Conservation scores 3 & 4 were calculated similarly as conservation score 2, after creation of multiple sequence alignments of the selected sequences.

**MCT8 homology modelling.** Structural models of the MCT8 protein were constructed using YASARA Structure, using previously described methods[47]. A novel hybrid MCT8 homology model was generated based on the Cryo-EM structures of MCT1 (Protein Data Bank [PDB]# 6LZ0, 7CKR, 6LYY) and MCT2 (PDB# 7BP3), as well as the crystal structure of the major facilitator superfamily protein FucP (PDB# 3O7Q) and bacterial MFS (PDB# 6HCL), using the homology modelling macro of YASARA Structure[58]. A T4 substrate molecule was docked at the substrate binding pocket, using the ligand docking tool implemented in YASARA, which is based on a derivative of Autodock[59], and the MCT8-T4 complex was embedded in a lipid bilayer using a

previously described approach[47]. The resulting model was compared with the MCT8 model derived from AlphaFold2[60].

All disease-causing missense variants identified through our international consortium and meta-analysis were modelled individually in the MCT8 structural model and their apparent impact was visualized using previously described methods[61]. Moreover, the ddG of each possible amino acid substitution was calculated using MAESTRO[62] and FoldX[63]. For this purpose, we used the MCT8 structure without membrane embedding, to reduce potential artefacts introduced during membrane embedding.

The following parameters were derived and used as independent structural features in establishing the variant prediction tool. First, the membrane interaction surface area was derived by computing the molecular surface between MCT8 and the lipid bilayer. Second, the substrate interaction surface area was derived by computing the molecular surface between the MCT8 protein and the T4 molecule modelled at the substrate binding center. Third, the surface accessible surface area was determined by computing the molecular surface between MCT8 and the surrounding water solvent. For this purpose the substrate molecule was deleted from the soup, allowing maximal exposure of the residues in the substrate channel to the surrounding water molecules. Fourth, the flexibility of each residue was determined by calculating the RMSD after running a molecular dynamic simulation of 1 ns using a YASARA 2 force field. Finally, the distance to the T4 substrate molecule was calculated for each residue.

Using a similar approach, a novel MCT8 structure in the inward-open conformation was modeled, based on the inward-open cryo-EM structures of MCT1 (PDB# 7CKO and 7DA5) and MCT2 (PDB# 7BP3), and the bacterial MFS (PDB#6HCL).

Substrate transition through the substrate pore was modeled using the morphing macro embedded in YASARA Structure.

All images were created using YASARA Structure and Pov-Ray v3.6 software (www.povray.org).

**MCT8 reference sequence.** The nomenclature of all MCT8 variants at cDNA and protein level were based on the following reference sequences, which represent the longest isoform of MCT8 mRNA and protein: NM_006517.3 (mRNA) and NP_006508.1 (protein).

**Dual pathogenicity and severity classifier for MCT8 variants.** The MCT8 Pathogenicity Classifier was built based on unsupervised learning, relying on conservation of proteins across species[19] and supervised machine learning, using a similar approach as described previously[64]. The two approaches were integrated by utilizing the outcomes of the unsupervised machine learning (evolutionary indices) as a feature in the supervised machine learning approach. One algorithm discriminated between benign and pathogenic variants, and a subsequent algorithm segregated predicted pathogenic variants in mildly pathogenic or combined moderately and severely pathogenic.

**Unsupervised Machine Learning.** Evolutionary indices were built following the principles and methods described in Frazer et al.[19], and were optimized for MCT8. Results were obtained by ensembling 10 random initialisations of the variational autoencoder.

**Supervised and Integrated Machine Learning**
**Feature engineering.** Impact of amino acid substitution: The impact of amino acid substitution was quantified using 91 substitution scoring matrices available in AAindex2 database[65].

Sequence-based scores: 4 different conservation scores were built using different sequence alignments (see section 'Conservation analyses').

Structural features: a total of 37 structural features were built, all of them were calculated using three different models: homology model in the inward-open conformation, homology model in the outward-open conformation (see section 'MCT8 homology modelling') and AlphaFold2 model[60]. Topology features were determined by visualization. Surface accessible areas were calculated using YASARA Structure (see section 'MCT8 homology modelling'). Center of Mass was calculated using Pymol. Changes in dG were calculated using MAESTRO and FoldX[66].

MCT8-specific variant classifier: A total of 480 validated MCT8 variants with 134 features, including evolutionary indices built as described in section 'Unsupervised Machine Learning', were used as the initial dataset to develop the two-steps classifier. The data was normalized using the Standard Scaler function of the pre-processing package of sckit-learn [Scikit-learn: Machine Learning in Python[67]]. Features were selected performing a Recursive Feature Elimination (RFE) using as model method a Logistic Regressor Algorithm and keeping the top ranked.

For the first classifier, 9 features were selected: LINK010101 score, GEOD900101 score, DAYM780301 score, Intracellular Loop residue, Extracellular Loop residue, Transmembrane domain residue, evolutionary index, FoldX ddG calculated on the homology model in the outward-open conformation, Mean distance to substrate calculated on the homology model in the outward-open conformation. A Logistic Regressor classifier to discriminate between Benign and Pathogenic MCT8 missense variants (Relative Activity cut-off = 50%) was developed using the scikit-learn library (Scikit-learn: Machine Learning in Python[67]). With 36% residual uptake as the highest residual T4 uptake in patient variants expressed in JEG3 cells, a 40% as upper limit was used in the alanine scanning (see below). An additional 10% was added to the Relative Activity cut-off, resulting in a cut-off of 50%, to limit the possibility to miss out potential novel patients with milder phenotypes. Hyperparameter tuning was performed using a Grid Search (optimizing the f1 score) obtaining the best parameters fitting the model: C = 1, penalty=l2 (Ridge), solver=saga; the remaining parameters were kept as the default model. Ten-fold cross-validation was used for assessing the performance of the model. To check if the model had merely 'memorized' highly conserved sites, we also evaluated the performance of the classifier by 10-fold cross-validation splitting the dataset by positions; we did not find relevant differences in performance metrics (see Supplementary Fig. 32).

For the second classifier, we only used those variants with relative activity below 50%. A total of 22 features were selected: Membrane Interacting Surface Area calculated on the homology model in the outward-open conformation, Substrate Channel Surface Area calculated on the homology model in the inward-open conformation, Membrane Interacting Surface Area calculated on the homology model in the inward-open conformation, Conservation Score across all MCT8 sequences, MEHP950101 score, OVEJ920103 score, FEND850101 score, LUTR910107 score, FITW660101 score, JOHM930101 score, MOHR870101 score, KOSJ950107, CSEM940101 score, DOSZ010103 score, KOSJ950101 score, GIAG010101 score, KOSJ950114 score, Residue in a Substrate Interaction Surface, evolutionary index, Transmembrane domain residue, Residue with Substrate Interaction, Residue in an alpha Helix. A Logistic Regressor classifier to discriminate between Moderate Pathogenic and Severe Pathogenic MCT8 missense variants (Relative Activity cut-off = 10%) was developed using the scikit-learn library. Hyperparameter tuning was performed using a Grid Search obtaining the best parameters fitting the model: C = 10, l1_ratio=0.4, penalty=elasticnet, solver=saga; the remaining parameters were kept as the default model. Ten-fold cross-validation was used for assessing the performance of the model. To check if the model had merely 'memorized' highly conserved sites, we also evaluated the performance of the classifier by 10-fold cross-validation splitting the dataset by positions; we did not find relevant differences in performance metrics (see Supplementary Fig. 32).

We evaluated feature relevance by performing a feature permutation importance inspection, and found evolutionary scores and amino acid properties being most important (Supplementary Fig. 33)).

Visualization: Figures were created using SeaBorn Python Library[68].

### Reporting summary

Further information on research design is available in the Nature Portfolio Reporting Summary linked to this article.

## Data availability

UK Biobank genotype and phenotype data on which several results of this study were based through the UK Biobank upon application and with permission of UKBB's Research Ethics Committee under accession codes 67864 and 27412 (http://www.ukbiobank.ac.uk/). The summary statistics from the GWAS meta-analyses used in this project are available on the ThyroidOmics Consortium website (http://www.thyroidomics.com) at the Datasets section (https://transfer.sysepi.medizin.uni-greifswald.de/thyroidomics/datasets/). The data of gnomAD (screening for non-synonymous variants in controls) used in this study are available in the gnomAD database (https://gnomad.broadinstitute.org/). The PDB files of the optimized MCT8 homology models are available at Zenodo (https://doi.org/10.5281/zenodo.14561250). The evolutionary indices generated in this study are available in the Source Data File. Raw sequencing data of patient variants that are in our possession are available on request from W.E. Visser (w.e.visser@erasmusmc.nl), with the period for response to the access request of one calendar month. Because of the rarity of MCT8 deficiency, individual participant data beyond that reported here will not be shared, to safeguard patient privacy following (local) data privacy law like the General Data Protection Regulation. Source data are provided as a Source Data file. Source data are provided with this paper.

## Code availability

The codebase for the EVE model is available at github.com/OATML-Markslab/EVE. The codebase for the MCT8 classifier is available at github: https://github.com/martin-mariano/MCT8classifier.

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

## Acknowledgements

Our study was funded by Eurostars (project number E11337; to W.E.V.), the Sherman Foundation (to W.E.V.). The centres in Rotterdam, Bucharest, Paris and Angers are part of the European Reference Network (ERN) on rare endocrine conditions (Endo-ERN). The centres in Rome and Amsterdam are members of the ERN for rare neurological disorders (ERN-RND) and the centres in Naples and Gdansk are members of ERN-ITHACA. The centre in Cambridge (UK) is supported by the Wellcome Trust (Investigator Award 210755/Z/18/Z to K.C.) and the National Institute of Health Research Cambridge Biomedical Research Centre. Funding by NIMH (funding number R01MH118281) to C.H.C). This research has been conducted using the UK Biobank Resource under application numbers 20272, 67864 and 27412. We thank the patients and their families and caregivers who participated in this study.

## Author contributions

M.M., M.D., J.F., C.M.M.G., R.B.T.M.S. and H.W. had equal contributions. S.G., F.v.G and W.E.V. conceived and designed the study and drafted the manuscript. S.G., F.S.v.G., A.D.C., L.J.d.R, S.L. and A.L.M. performed experimental studies. M.M., M.D., J.F.,D.S.M. and J.P.N. conducted or oversaw machine-learning studies. C.M.M.G., R.B.T.M.S., H.W., A.T., F.R., C.H.C. and M.M. conducted or oversaw genetic studies. S.G., F.S.v.G., A.A., E.L.T.v.d.A., G.P.A., C.M.A., I.B., P.B., D.B., A.J.B., S.A.A.v.d.B., A.v.d.B., E.B., I.M.v.B., N.B.P., D.B., M.C., G.C., B.C., C.C., K.C., A.C., P.C., J.C.v.d.S., I.F.M.d.C., R.C., D.C., P.C., C.D., K.D., C.D., A.D., P.D., M.H.G.D., R.D., A.E., J.F., J.G., L.G., B.G., E.F.G., E.G., A.H., Z.H., B.H., A.C.H., T.H., A.R.I., A.K., M.M.v.d.K., D.K., D.A.K., H.K., A.K., A.L., S.H.L., A.L.Y., J.L., M.L.L., C.F.L., C.M.L., R.J.L., G.L., J.M., E.E.M., K.L.M., A.M., V.M., F.M.L., C.M., K.E.M., L.E.N., I.O.P., L.P., P.G.P., M.P., F.P., F.O.P., C.R., K.R., R.S., T.S.M., P.S., A.S., Y.S., M.S., M.A.M.S., M.T.S., A.S., G.M.S., L.S., D.T., S.T., J.V., A.v.d.W., J.L.W., M.v.W., J.W., M.C.Y.d.W., N.I.W., M.W., F.Z., A.Z., N.Z.S.,W.E.V. contributed information on patients. All authors (S.G., F.S.v.G., M.M., M.D., J.F., M.C.M.M.G., R.B.T.M.S., H.W., A.D.C., L.J.d.R., A.T., A.A., E.L.T.v.d.A., G.P.A., C.M.A., I.B., P.B., D.B., A.J.B., S.A.A.v.d.B., A.v.d.B., E.B., I.M.v.B., N.B.P., D.B., M.C., G.C., B.C., C.C., K.C., A.C., P.C., J.C.v.d.S., I.F.M.d.C., R.C., D.C., P.C., C.D., K.D., C.D., A.D., P.D., M.H.G.D., R.D., A.E., J.F., J.G., L.G., B.G., E.F.G., E.G., A.H., Z.H., B.H., A.C.H., T.H., A.R.I., A.K., M.M.v.d.K., D.K., D.A.K., H.K., A.K., A.L., S.H.L., A.L.Y., J.L., S.L., M.L.L., A.L.M., C.F.L., C.M.L., R.J.L., G.L., J.M., E.E.M., K.L.M., A.M., V.M., F.M.L., C.M., K.E.M., L.E.N., I.O.P., L.P., P.G.P., M.P., F.P., F.O.P., C.R., K.R., R.S., T.S.M., P.S., A.S., Y.S., M.S., M.A.M.S., M.T.S., A.S., G.M.S., L.S., D.T., S.T., J.V., A.v.d.W., J.L.W., M.v.W., J.W., M.C.Y.d.W., N.I.W., M.W., F.Z., A.Z., N.Z.S., F.R., M.E.M., D.S.M., J.P.N., C.H.C., M.M., W.E.V.) had a critical review of the manuscript.

## Competing interests

The Erasmus Medical Center (Rotterdam, Netherlands), which employs SG, FSvG, RBTMS, ELTvdA, SAAvdB, IMvB, MHGD, SL, ALM, MTS, HvT, MdW, RvdW, MCYdW, MEM, MM and WEV receives royalties from Egetis Therapeutics (the manufacturer of Triac). None of the authors will benefit personally from any royalties. Egetis Therapeutics had no influence on the conduct or analysis of this study. The remaining authors declare no competing interests.

## Additional information

Stefan Groeneweg[1,96], Ferdy S. van Geest [1,96], Mariano Martín[2,3], Mafalda Dias [4,5], Jonathan Frazer[4,5], Carolina Medina-Gomez [6], Rosalie B. T. M. Sterenborg [1,7], Hao Wang [8], Anna Dolcetta-Capuzzo[1], Linda J. de Rooij[1], Alexander Teumer [9,10], Ayhan Abaci[11], Erica L. T. van den Akker[12], Gautam P. Ambegaonkar[13], Christine M. Armour[14], Iuliu Bacos[15], Priyanka Bakhtiani[16,91], Diana Barca[17], Andrew J. Bauer [18], Sjoerd A. A. van den Berg[19], Amanda van den Berge[1], Enrico Bertini [20], Ingrid M. van Beynum[21], Nicola Brunetti-Pierri [22,23,24], Doris Brunner[25], Marco Cappa[26], Gerarda Cappuccio[22,27], Barbara Castellotti[28], Claudia Castiglioni [29], Krishna Chatterjee [30], Alexander Chesover[31,92], Peter Christian [32], Jet Coenen-van der Spek [33], Irenaeus F. M. de Coo[34], Regis Coutant[35], Dana Craiu[17], Patricia Crock[36], Christian DeGoede [37], Korcan Demir [11], Cheyenne Dewey[38], Alice Dica[17], Paul Dimitri[39], Marjolein H. G. Dremmen[40], Rachana Dubey[41], Anina Enderli[42], Jan Fairchild[43], Jonathan Gallichan[44], Luigi Garibaldi[45], Belinda George[46], Evelien F. Gevers[47], Erin Greenup [48,93], Annette Hackenberg[42], Zita Halász[49], Bianka Heinrich[42], Anna C. Hurst [50], Tony Huynh [51], Amber R. Isaza[18], Anna Klosowska[52], Marieke M. van der Knoop[53], Daniel Konrad [54], David A. Koolen [33], Heiko Krude[55], Abhishek Kulkarni[56], Alexander Laemmle [57], Stephen H. LaFranchi[58], Amy Lawson-Yuen [38,59], Jan Lebl [60], Selmar Leeuwenburgh[1], Michaela Linder-Lucht[61], Anna López Martí[1], Cláudia F. Lorea[62,94], Charles M. Lourenço[63,64], Roelineke J. Lunsing[65], Greta Lyons[30], Jana Krenek Malikova[60], Edna E. Mancilla[18],

# Article

Kenneth L. McCormick [48], Anne McGowan[30], Veronica Mericq[66], Felipe Monti Lora[67], Carla Moran [30], Katalin E. Muller [68], Lindsey E. Nicol[58], Isabelle Oliver-Petit[69], Laura Paone[70], Praveen G. Paul[71], Michel Polak[72], Francesco Porta[73], Fabiano O. Poswar [74], Christina Reinauer[75], Klara Rozenkova[60], Rowen Seckold[36], Tuba Seven Menevse[76], Peter Simm[77], Anna Simon[71], Yogen Singh [78,95], Marco Spada[73], Milou A. M. Stals[1], Merel T. Stegenga [1], Athanasia Stoupa[72], Gopinath M. Subramanian[36], Lilla Szeifert[49], Davide Tonduti[79,80], Serap Turan [76], Joel Vanderniet[36], Adri van der Walt[81], Jean-Louis Wémeau[82], Anne-Marie van Wermeskerken[83], Jolanta Wierzba[84], Marie-Claire Y. de Wit [53], Nicole I. Wolf [85,86], Michael Wurm[87], Federica Zibordi[88], Amnon Zung[89], Nitash Zwaveling-Soonawala[90], Fernando Rivadeneira [6], Marcel E. Meima [1], Debora S. Marks [4], Juan P. Nicola[2], Chi-Hua Chen [8], Marco Medici [1] & W. Edward Visser [1] ✉

[1]Academic Center for Thyroid Diseases, Department of Internal Medicine, Erasmus Medical Centre, Rotterdam, The Netherlands. [2]Department of Clinical Biochemistry (CIBICI-CONICET), Faculty of Chemical Sciences, National University of Córdoba, Córdoba, Argentina. [3]Institute for Bioengineering of Catalonia (IBEC), The Barcelona Institute of Science and Technology, Barcelona, Spain. [4]Department of Systems Biology, Harvard Medical School, Boston, MA, USA. [5]Centre for Genomic Regulation (CRG), The Barcelona Institute of Science and Technology, Barcelona, Spain Universitat Pompeu Fabra (UPF), Barcelona, Spain. [6]Department of Internal Medicine, Erasmus University Medical Center, Rotterdam, The Netherlands. [7]Department of Internal Medicine, Division of Endocrinology, Radboud University Medical Center, Nijmegen, The Netherlands. [8]Center for Multimodal Imaging and Genetics, University of California San Diego, La Jolla, CA, USA. [9]Department of Psychiatry and Psychotherapy, University Medicine Greifswald, Greifswald, Germany. [10]DZHK (German Centre for Cardiovascular Research), Partner Site Greifswald, Greifswald, Germany. [11]Division of Pediatric Endocrinology, Faculty of Medicine, Dokuz Eylul University, İzmir, Turkey. [12]Department of Paediatrics, Division of Endocrinology, Erasmus Medical Centre -Sophia Children's Hospital, Rotterdam, The Netherlands. [13]Department of Paediatric Neurology, Addenbrooke's Hospital, Cambridge University Hospitals NHS Foundation Trust, Cambridge, UK. [14]Regional Genetics Program, Children's Hospital of Eastern Ontario and Children's Hospital of Eastern Ontario Research Institute, University of Ottawa, Ottawa, ON, Canada. [15]Centrul Medical Dr. Bacos Cosma, Timisoara, Romania. [16]University of Louisville, Louisville, KY, USA. [17]Carol Davila University of Medicine, Department of Clinical Neurosciences, Paediatric Neurology Discipline II, Bucharest, Romania. [18]Division of Endocrinology and Diabetes, Children's Hospital of Philadelphia, Philadelphia, PA, USA. [19]Diagnostic Laboratory for Endocrinology, Department of Internal Medicine, Erasmus Medical Center, Rotterdam, The Netherlands. [20]Unit of Neuromuscular and Neurodegenerative Disorders, Bambino Gesu' Children's Research Hospital IRCCS, Rome, Italy. [21]Department of Pediatrics, Division of Pediatric Cardiology, Erasmus Medical Centre - Sophia Children's Hospital, Rotterdam, The Netherlands. [22]Department of Translational Medicine, Federico II University, 80131 Naples, Italy. [23]Telethon Institute of Genetics and Medicine (TIGEM), Pozzuoli, Naples, Italy. [24]Scuola Superiore Meridionale (SSM, School of Advanced Studies), Genomics and Experimental Medicine Program, University of Naples Federico II, Naples, Italy. [25]Gottfried Preyer's Children Hospital, Vienna, Austria. [26]Research Area for Innovative Therapies in Endocrinopathies, Bambino Gesù Children's Hospital, IRCCS, Rome, Italy. [27]Neurological Research Institute and Baylor College of Medicine, Houston, TX, USA. [28]Unit of Medical Genetics and Neurogenetics, Fondazione IRCCS Istituto Neurologico Carlo Besta, Milan, Italy. [29]Department of Neurology, Clinica Meds, School of Medicine, Universidad Finis Terrae, Santiago, Chile. [30]Wellcome Trust-Medical Research Council Institute of Metabolic Science, University of Cambridge, Cambridge, UK. [31]Division of Endocrinology, The Hospital for Sick Children and Department of Paediatrics, University of Toronto, Toronto M5G 1×8, Canada. [32]East Kent Hospitals University NHS Foundation Trust, Ashford, UK. [33]Department of Human Genetics, Donders Institute for Brain, Cognition and Behaviour, Radboud University Medical Center (Radboudumc), Nijmegen, The Netherlands. [34]Department of Toxicogenomics, Unit Clinical Genomics, Maastricht University, MHeNs School for Mental Health and Neuroscience, Maastricht, The Netherlands. [35]Department of Pediatric Endocrinology and Diabetology, University Hospital, Angers, France. [36]John Hunter Children's Hospital, Hunter Medical Research Institute, University of Newcastle, New Lambton Heights, Australia. [37]Department of Paediatric Neurology, Clinical Research Facility, Lancashire Teaching Hospitals NHS Trust, Lancashire, UK. [38]Genomics Institute Mary Bridge Children's Hospital, MultiCare Health System, Tacoma, WA, USA. [39]The Department of Oncology and Metabolism, The University of Sheffield, Western Bank, Sheffield S10, 2TH, UK. [40]Division of Paediatric Radiology, Erasmus Medical Centre – Sophia's Children Hospital, Rotterdam, The Netherlands. [41]Medanta Superspeciality Hospital, Indore, India. [42]Department of Neuropediatrics, University Children's Hospital, University of Zurich, Zurich, Switzerland. [43]Department of Diabetes and Endocrinology, Women's and Children's Hospital, North Adelaide 5066 South Australia, Australia. [44]Plymouth Hospitals NHS Trust, Plymouth, UK. [45]UPMC Children's Hospital of Pittsburgh, Pittsburgh, PA, USA. [46]Department of Endocrinology, St. John's Medical College Hospital, Bengaluru, India. [47]Centre for Endocrinology, William Harvey Research institute, Queen Mary University of London, London, UK. [48]Department of Pediatrics, Division of Pediatric Endocrinology, University of Alabama at Birmingham, Birmingham, AL, USA. [49]Pediatric Center, Semmelweis University Budapest, Budapest, Hungary. [50]Department of Genetics, University of Alabama at Birmingham, Birmingham, AL, USA. [51]Department of Endocrinology & Diabetes, Queensland Children's Hospital, South Brisbane, Queensland, Australia. [52]Department of Pediatrics, Hematology and Oncology, Medical University of Gdańsk, Gdańsk, Poland. [53]Department of Paediatric Neurology, Erasmus Medical Centre, Rotterdam, The Netherlands. [54]Division of Pediatric Endocrinology and Diabetology and Children's Research Center, University Children's Hospital, University of Zurich, Zurich, Switzerland. [55]Institute of Experimental Paediatric Endocrinology, Charité-Universitätsmedizin Berlin, Berlin, Germany. [56]Department of Paediatric Endocrinology, SRCC Children's Hospital, Mumbai, India. [57]Institute of Clinical Chemistry and Department of Pediatrics, Inselspital, University Hospital Bern, Bern, Switzerland. [58]Department of Pediatrics, Doernbecher Children's Hospital, Oregon Health & Sciences University, Portland, OR, USA. [59]Department of Genetics, Kaiser Permanente Washington, Seattle, WA, USA. [60]Department of Paediatrics, Second Faculty of Medicine, Charles University, University Hospital Motol, Prague, Czech Republic. [61]Division of Neuropediatrics and Muscular Disorders, Department of Pediatrics and Adolescent Medicine, University Hospital Freiburg, Freiburg, Germany. [62]Teaching Hospital of Universidade Federal de Pelotas, Pelotas, Brazil. [63]National Reference Center for Rare Diseases, Faculdade de Medicina de São José do Rio Preto, São José do Rio Preto, Brazil. [64]Personalized Medicine area -Special Education Sector at DLE/Grupo Pardini, Rio de Janeiro, Brazil. [65]Department of Child Neurology, University Medical Center Groningen, University of Groningen, Groningen, The Netherlands. [66]Institute of Maternal and Child Research, University of Chile, Santiago, Chile, Department of Pediatrics, Clinica Las Condes, Santiago, Chile. [67]Pediatric Endocrinology Group, Sabara Children's Hospital, São Paulo, Brazil. [68]Heim Pal National Pediatric Institute, Budapest, Hungary. [69]Department of Paediatric Endocrinology and Genetics, Children's Hospital, Toulouse University Hospital, Toulouse, France. [70]Endocrinology and Diabetology Unit, Bambino Gesù Children's Hospital, IRCCS, Rome, Italy. [71]Department of Paediatrics, Christian Medical College, Vellore, India. [72]Paediatric Endocrinology, Diabetology and Gynaecology, Department, Necker Children's University Hospital, Imagine Institute Affiliate, Université de Paris Cité, Paris, France. [73]Department of Paediatrics, AOU Città della Salute e della Scienza di Torino, University of Torino, Turin, Italy. [74]Medical Genetics Service, Hospital de Clínicas de Porto Alegre, Porto Alegre, Brazil. [75]Department of General Pediatrics, Neonatology

and Pediatric Cardiology, University Children's Hospital, Medical Faculty, Dusseldorf, Germany. [76]Marmara University School of Medicine Department of Pediatric Endocrinology, Istanbul, Turkey. [77]Royal Children's Hospital/University of Melbourne, Parkville, Australia. [78]Department of Paediatric Cardiology, Addenbrooke's Hospital, Cambridge University Hospitals NHS Foundation Trust, Cambridge, UK. [79]Child Neurology Unit - C.O.A.L.A. (Center for diagnosis and treatment of leukodystrophies), V. Buzzi Children's Hospital, Milano, Italy. [80]Department of Clinical and Biomedical Science, Università degli Studi di Milano, Milano, Italy. [81]Private paediatric Neurology practice Dr A van der Walt, Durbanville, South Africa. [82]University of Lille, Lille, France. [83]Department of Paediatrics, Flevoziekenhuis, Almere, The Netherlands. [84]Department of Internal and Pediatric Nursing, Institute of Nursing and Midwifery, Medical University of Gdańsk, Gdańsk, Poland. [85]Department of Child Neurology, Amsterdam Leukodystrophy Center, Emma Children's Hospital, Amsterdam University Medical Centers, Vrije Universiteit Amsterdam, Amsterdam, The Netherlands. [86]Amsterdam Neuroscience, Cellular & Molecular Mechanisms, Amsterdam, The Netherlands. [87]University Children's Hospital Regensburg (KUNO), University of Regensburg, Campus St. Hedwig, Regensburg, Germany. [88]Child Neurology Unit, Fondazione IRCCS, Istituto Neurologico Carlo Besta, Milan, Italy. [89]Pediatric Endocrinology Unit, Kaplan Medical center, Rehovot and the Hebrew University of Jerusalem, Jerusalem, Israel. [90]Emma Children's Hospital, Department of Paediatric Endocrinology, Amsterdam UMC, University of Amsterdam, Amsterdam, The Netherlands. [91]Present address: Childrens Hospital Los Angeles, Los Angeles, CA, USA. [92]Present address: Department of Endocrinology, Great Ormond Street Hospital for Children, London, UK. [93]Present address: Division of Pediatric Endocrinology, Department of Pediatrics, Orlando Health Arnold Palmer Hospital for Children, Orlando, FL, USA. [94]Present address: Federal University of Rio Grande do Sul, Porto Alegre-RS, Brazil. [95]Present address: Department of Pediatrics, University of California - UC Davis Children's Hospital, Sacramento, CA, USA. [96]These authors contributed equally: Stefan Groeneweg, Ferdy S. van Geest. ✉e-mail: w.e.visser@erasmusmc.nl

