## [Peer Review file · Nature Communications]

Mapping variants in thyroid hormone transporter MCT8 to disease severity by genomic, phenotypic, functional, structural and deep learning integration

Corresponding Author: Professor W. Edward Visser

Version 0:

Reviewer comments:

Reviewer #1

(Remarks to the Author)

This is a well designed study evaluating genotype-phenotype correlations in most, if not all, patients presently known to be affected with MCT8 defects. The Authors applied genetic, molecular, structural, biochemical and clinical studies to understand and the pathogenic impact of the identified variants. The work included the investigation of variants identified in the general population and provides the MCT8 mutation atlas which is claimed by the Authors to become useful for the potential development of novel therapeutic approaches. The manuscript is well written and the Authors acknowledge the major limitations of their study but the significance of their findings is nevertheless solid. These results can become useful also for other MCT-related disorders.

I have some specific comments:

- this is an X-linked disease affecting mainly males. This should be clearly stated for the general readers of the journal;
- whatever the type of treatment that will be developed in the future, its effectiveness will be higher if this could be started during the early phases of neurological development, therefore a major effort should be made to identify clinical/biochemical manifestations allowing the recognition of heterozygous female carriers before the birth of an affected male in the family. An attempt to obtain insights into the the female relatives (with or without skewed X chr. inactivation) of the affected patients appears feasible and would be of great value.

Reviewer #2

(Remarks to the Author)

The manuscript entitled „Mapping variants in thyroid hormone transporter MCT8 to disease severity by genomic, phenotypic, functional, structural and deep learning integration” by Groeneweg and Geest et al assembles a large cohort of individuals affected by MCT8 mutations with the goal to align phenotype and genotype in a comprehensive manner. This is complemented by a cohort with TRIAC therapy, which adds a novel aspect to the discussion. Overall the study is impressive in quality and quantity, and the clinical and molecular parts are well done and coherent.

Minor comments:

1. The authors focus on male MCT8 deficiency, which is understandable given the x-linkage. However, given the increasing gender discussion and the demands of Nat Comm regarding this aspect, I would appreciate if a few words could be added to the discussion, ie whether (compound) homozygous KO females exist (given that affected males cannot reproduce, presumably extremely rare) and whether heterozygous females could also be affected given the x-inactivation.
2. For people not in the field it would be good to add one sentence what Ala-scanning means and why it is relevant for the approach.
3. The authors use 3-letter and 1-letter codes (line 334 and 341). Please use consistently only one.
4. Table 1 summarizes the key clinical features. However, in addition to the occurrence of a phenotype it would be nice to indicate how much it was altered in those cases (e.g. hypertension or QTc) in average.

Reviewer #3

(Remarks to the Author)

The authors systematically gathered genetic, clinical, and biochemical data from individuals with MCT8 deficiency. These data were also integrated with molecular studies, extensive alanine-scanning of the protein, and in silico predictions to understand the fundamental molecular mechanisms of MCT8 deficiency. These efforts pave the way for future research on MCT8 deficiency.

However, I have several concerns regarding the computational models used for predicting the pathogenicity and severity of MCT8 variants:

1. Several key training details are missing. For instance, during cross-validation, was the dataset split by variants or by positions? I suggest splitting by positions to prevent the model from merely 'memorizing' highly conserved sites.
2. How do other recent state-of-the-art computational methods, such as gMVP (Zhang et al., Nature Machine Intelligence, 2022) and AlphaMissense (Cheng et al., Science, 2023), perform in predicting the pathogenicity of MCT8 variants?
3. Both EVE and language model-based methods, such as ESM-1v (Meier et al., NeurIPS, 2021), are unsupervised methods for predicting variant effects and often complement each other. Can the performance of ESM-1v on pathogenicity and severity prediction also be demonstrated?
4. What are the most important features for pathogenicity and severity prediction, respectively? Ablation studies on features would be helpful in understanding how the computational methods work.
5. Precision-recall curve is another crucial metric for evaluating performance, particularly when the number of positive and negative samples is highly imbalanced. In practice, we usually prioritize prediction accuracy under a specific threshold of recall rate. Could you also benchmark the performance using a precision-recall curve and calculate the area under the precision-recall curve (auPRC)?

Version 1:

Reviewer comments:

Reviewer #1

(Remarks to the Author)

The Authors responded adequately to the comments. The manuscript has been further improved.

Reviewer #2

(Remarks to the Author)

The authors have addressed my comments satisfyingly.

Reviewer #3

(Remarks to the Author)

Thanks for your response. All my concerns have been addressed.

REVIEWER COMMENTS

Reviewer #1 (Remarks to the Author):

This is a well designed study evaluating genotype-phenotype correlations in most, if not all, patients presently known to be affected with MCT8 defects. The Authors applied genetic, molecular, structural, biochemical and clinical studies to understand and the pathogenic impact of the identified variants. The work included the investigation of variants identified in the general population and provides the MCT8 mutation atlas which is claimed by the Authors to become useful for the potential development of novel therapeutic approaches. The manuscript is well written and the Authors acknowledge the major limitations of their study but the significance of their findings is nevertheless solid. These results can become useful also for other MCT-related disorders.

>> We thank the reviewer for the positive evaluation of our manuscript and the comments raised.

I have some specific comments:

- this is an X-linked disease affecting mainly males. This should be clearly stated for the general readers of the journal;

>> We agree with the reviewer and have added the words "located at the X-chromosome" to the abstract (page 8, line 198) and introduction (page 9, line 225).

- whatever the type of treatment that will be developed in the future, its effectiveness will be higher if this could be started during the early phases of neurological development, therefore a major effort should be made to identify clinical/biochemical manifestations allowing the recognition of heterozygous female carriers before the birth of an affected male in the family. An attempt to obtain insights into the the female relatives (with or without skewed X chr. inactivation) of the affected patients appears feasible and would be of great value.

>> We fully agree that early diagnosis would enable immediate intervention, which has the greatest potential to achieve positive effects on brain development. Previously, in a world-wide deep-phenotyping effort, we found that pregnancy duration was normal and pregnancy were typically uneventful (Groeneweg, Lancet Diab Endo 2020). Also, Apgar scores >8 were seen in 94% of babies with MCT8 deficiency; moreover, they had normal body weight (median 3584 grams). This is compatible with reports of mothers from our MCT8 registry indicating that no consistent abnormalities were noted in prenatal ultrasounds (unpublished data; see also Van Geest, JCEM 2023).

We also agree with the relevance of improving insights on females with variants in MCT8. In fact, we have recently formulated a manuscript on females with mutations in MCT8. Over the last 6 years, we have collected clinical and biochemical data (n=33) as well as patient cells from females carrying mutations in MCT8. In those with unfavorable X-inactivation (n=7), we have found varying clinical features ranging from mild social impairments to a phenotype resembling MCT8 deficiency in males. This research project allowed us to compare thyroid function tests from non-carriers, carriers and male patients with MCT8 deficiency. This Figure (Rebuttal Fig. 1), shown

below, shows that only mean FT4 concentrations are somewhat lower in carriers vs non-carriers, but largely within the reference range. All other thyroid function tests (TSH, T3, rT3 and all ratios) in carriers were within the reference ranges. Together, this precludes such tests as markers to diagnose carriers. As mentioned, we intend to submit this manuscript with clinical, biochemical and molecular data (including patient-derived cells) from female carriers within the upcoming month. That manuscript includes additional analysis and differs substantially in scope from the current manuscript.

As we feel that incorporating such data into the present manuscript is beyond its scope, we hope that our intention to submit such data as the already formulated manuscript would be agreeable with the reviewer and editor. To accommodate the reviewer's remark, we have included a statement that the utilization of thyroid function tests to identify female carriers, with the aim of initiating potential antenatal therapies in affected fetuses has not been found to be feasible due to the lack of significant differences compared to non-carriers (unpublished observations, page 19, line 459-461). We have further specified in the discussion of the revised manuscript which types of early diagnostics (neonatal screening and prenatal testing) could be helpful in this disorder (page 19, line 456).

Rebuttal Figure 1. (Figure 4 in a manuscript entitled: MCT8 deficiency in females). Serum thyroid function tests in female patients in comparison to asymptomatic carriers and non-carriers. (A) serum free T4, (B) total T3, and (C) total rT3 concentrations, as well as (D) total T3/rT3 and (E) total T3/ free T4 ratios, and serum TSH concentrations in female patients (P), self-reported a-

symptomatic female carriers (C) and non-carriers (NC). One-way ANOVA with Tuckey`s post-hoc tests were used to test for statistically significant differences between groups (ns, not significant; $p < 0.05$, *; $p < 0.005$, **; $p < 0.0005$, ***; $p < 0.0001$, ****).

Reviewer #2 (Remarks to the Author):

The manuscript entitled „Mapping variants in thyroid hormone transporter MCT8 to disease severity by genomic, phenotypic, functional, structural and deep learning integration” by Groeneweg and Geest et al assembles a large cohort of individuals affected by MCT8 mutations with the goal to align phenotype and genotype in a comprehensive manner. This is complemented by a cohort with TRIAC therapy, which adds a novel aspect to the discussion. Overall the study is impressive in quality and quantity, and the clinical and molecular parts are well done and coherent.

>> We thank the reviewer for the positive evaluation of our manuscript and the comments raised.

Minor comments:

1. The authors focus on male MCT8 deficiency, which is understandable given the x-linkage. However, given the increasing gender discussion and the demands of Nat Comm regarding this aspect, I would appreciate if a few words could be added to the discussion, ie whether (compound) homozygous KO females exist (given that affected males cannot reproduce, presumably extremely rare) and whether heterozygous females could also be affected given the x-inactivation.

>> We agree that it is relevant, also in this manuscript, to mention the existence of phenotypes in females with mutations in MCT8 and unfavorable X-inactivation. We have added this in the discussion of the revised manuscript (page 18, lines 450-455).

We fully acknowledge the relevance of females with MCT8 deficiency, as also raised by reviewer #1. Therefore, we recently formulated a manuscript on females with mutations in MCT8. Over the last 6 years, we have collected clinical and biochemical data (n=33) as well as patient cells from females carrying mutations in MCT8. We plan to submit this manuscript with clinical, biochemical and molecular data (including patient-derived cells) from female carriers within the upcoming month.

2. For people not in the field it would be good to add one sentence what Ala-scanning means and why it is relevant for the approach.

>> We have added a brief explanation in the main text of the manuscript (page 13, line 314-315).

3. The authors use 3-letter and 1-letter codes (line 334 and 341). Please use consistently only one.

>> We agree with the reviewer that this is more intuitive; also, using 1-letter codes are being used in the Nature journals. Therefore, we have changed all amino acid codes to a 1-letter code when mentioned in conjunction with a position.

4. Table 1 summarizes the key clinical features. However, in addition to the occurrence of a phenotype it would be nice to indicate how much it was altered in those cases (e.g. hypertension or QTc) in average.

>> We have added body weight, body height, head circumference, percentiles of systolic and diastolic blood pressure and QTc to Table 1 of the revised version of the manuscript.

Reviewer #3 (Remarks to the Author):

The authors systematically gathered genetic, clinical, and biochemical data from individuals with MCT8 deficiency. These data were also integrated with molecular studies, extensive alanine-scanning of the protein, and in silico predictions to understand the fundamental molecular mechanisms of MCT8 deficiency. These efforts pave the way for future research on MCT8 deficiency.

>> We thank the reviewer for the constructive comments which helped us to improve the manuscript.

However, I have several concerns regarding the computational models used for predicting the pathogenicity and severity of MCT8 variants:

1. Several key training details are missing. For instance, during cross-validation, was the dataset split by variants or by positions? I suggest splitting by positions to prevent the model from merely 'memorizing' highly conserved sites.

>> In the first version of the submitted manuscript we evaluated the classifier by 10-fold cross-validation, performing a random test-train splitting of the variants. We agree with the reviewer, and to avoid data-leakage regarding conserved sites at testing time, we also evaluated the performance of the classifier by 10-fold cross-validation now splitting the dataset by positions. The difference in performance metrics is marginal. When splitting by variants ROC AUCs are 0.91 ± 0.08 for the first step (Pathogenic vs Benign) and 0.86 ± 0.12 for the second step of the classifier (Severity prediction); when splitting by positions AUCs are 0.92 ± 0.06 for the first step and an $AUC = 0.89 \pm 0.1$ for the second step. These results revealed that our model hasn't "memorized" highly conserved sites.

ROC curves are provided as Supplementary Figures 32a and 32b, and we clarified this in the Supplementary Text manuscript (page 19).

2. How do other recent state-of-the-art computational methods, such as gMVP (Zhang et al., Nature Machine Intelligence, 2022) and AlphaMissense (Cheng et al., Science, 2023), perform in predicting the pathogenicity of MCT8 variants?

>> We compared the performance of our classifier with recent state-of-the-art computational methods. In step 1 (pathogenicity classifier), our model outperformed the recent state-of-the-art predictor ESM-1v (AUROC: 0.91 vs 0.87, p-value = 0.006161; AUPRC: 0.82 vs 0.71). Although our model did not significantly outperform AlphaMissense as measured by AUC (AUC 0.9; p=0.27), we noticed a substantial difference in terms of precision-recall, as indicated by the AUPRC (0.82 vs 0.74). AlphaMissense seems to have been optimized for maximizing recall and it is overpredicting pathogenicity. Specifically, from the initial dataset of 480 variants where 106 variants are pathogenic, Alphamissense predicts 243 variants to be pathogenic. gMVP pre-scored dataset only contains SNVs missense variants and, unfortunately, not all possible missense variants as stated in the paper. From our dataset of 480 variants, only 237 have a gMVP score. Our classifier (AUC=0.94) outperforms gMVP (AUC=0.91) (DeLong’s test p-value=0.02796) in this subset of variants. In step 2 (severity classifier), our classifier outperformed all the state-of-the-art predictors, both in AUROC and AUPRC (MCT8-classifier: AUROC = 0.87, AUPRC = 0.92; AlphaMissense: AUROC = 0.71, AUPRC =0.80, ESM-1v: AUROC = 0.71, AUPRC =0.81, EVE: AUROC = 0.7, AUPRC =0.79, PolyPhen: AUROC =0.62, AUPRC =0.74 and SIFT: AUROC =0.55, AUPRC =0.72; gMVP subset MCT8-classifier AUC=0.89, gMVP AUROC = 0.7; DeLong’s test p-value<0.01 for all the comparisons).

For the reasons mentioned, the incomplete gMVP scores precluded a proper comparison with the other predictors. Therefore, we decided to add only AlphaMissense and ESM-1v to Figure 4c and 4d in the revised manuscript. For clarity, we have added the Figure containing data from the gMVP subset as Figure 2 (see below) in this rebuttal letter. Should the reviewer and editor believe it is useful to the readership, we would be happy to include this as a Supplementary Figure.

Rebuttal Figure 2. Performance of the MCT8 pathogenicity (left) and severity (right) classifier for all functionally evaluated variants as shown by AUC using the gMVP prediction tool.

We added clarification (page 15, lines 360-363 and lines 366-367) and added them to Figures 4c and 4d and Supplementary Figure 28.

3. Both EVE and language model-based methods, such as ESM-1v (Meier et al., NeurIPS, 2021), are unsupervised methods for predicting variant effects and often complement each other. Can the performance of ESM-1v on pathogenicity and severity prediction also be demonstrated?

>> ESM-1v performance on predicting MCT8 pathogenicity and severity was evaluated as for EVE and other-state-of-the-art classifiers (see response to comment 2). ESM-1v has a similar performance to EVE in pathogenicity and severity prediction. Our new MCT8-specific classifier outperforms both predictors in both predictions (DeLong's test: first step p-value=0.006161; second step p-value=0.003199).

We added clarification on page 15 (lines 360-363 and lines 366-367) and added this to Figures 4c and 4d) and Supplementary Figure 28).

4. What are the most important features for pathogenicity and severity prediction, respectively? Ablation studies on features would be helpful in understanding how the computational methods work.

>> We evaluated feature relevance by performing a feature permutation importance inspection. For the first step of the classifier, the most relevant feature is EVE score followed by amino acid index LINK010101 (Lin et al., 2001) and GEOD900101 (George et al., 1990) and ddG scores calculated by FoldX in the AlphaFold model in the outward open conformation. For the second step of the classifier, the most relevant feature is the amino acid index MOHR870101 (Mohana Rao, 1987), this is a scoring matrix based on physical parameters. In second place the amino acid index KOSJ950101 (Koshi-Goldstein, 1995) is found and in the third place the EVE score. Full feature-importance bar plots are provided as Supplementary Figures 33a and 33b and reference to in the Supplementary Methods (page 20).

5. Precision-recall curve is another crucial metric for evaluating performance, particularly when the number of positive and negative samples is highly imbalanced. In practice, we usually prioritize prediction accuracy under a specific threshold of recall rate. Could you also benchmark the performance using a precision-recall curve and calculate the area under the precision-recall curve (auPRC)?

>> As suggested by this reviewer, and as already discussed in our response to comment 2 above, we also evaluated the performance of our predictor by precision-recall curves.

The first step of our classifier outperforms all other models based on AUPRC (0.82 vs AlphaMissense: 0.74, ESM-1v: 0.71, EVE: 0.76, PolyPhen:0.64 and SIFT:0.51). Our classifier also has higher recall rate at any given threshold than other predictors, except for a precision between 50-60% where AlphaMissense presents a higher recall rate.

The second step of our classifier outperforms all the state-of-the-art predictors as showed in PRC curves and by auPRC, in predicting the severity of pathogenic mutations (0.92 vs AlphaMissense: 0.8, ESM-1v: 0.81, EVE: 0.79, PolyPhen:0.74 and SIFT:0.72). PRC curves are provided in Supplementary Figure 28.

To make this comment insightful for the readership, we also referenced to those Suppl Figures in the legends of Fig. 4c and 4d.

We mention about the precision-recall and AUPRC in the revised version of the manuscript (page 15, line 362-363).